# A medullary centre for lapping in mice

Bowen Dempsey[1], Selvee Sungeelee[1], Phillip Bokiniec [2], Zoubida Chettouh[1], Séverine Diem [3], Sandra Autran[3], Evan R. Harrell[4], James F. A. Poulet [2], Carmen Birchmeier[5], Harry Carey[6], Auguste Genovesio[1], Simon McMullan[6], Christo Goridis[1], Gilles Fortin[1,7] & Jean-François Brunet [1,7 ✉]

It has long been known that orofacial movements for feeding can be triggered, coordinated, and often rhythmically organized at the level of the brainstem, without input from higher centers. We uncover two nuclei that can organize the movements for ingesting fluids in mice. These neuronal groups, IRt$^{Phox2b}$ and Peri5$^{Atoh1}$, are marked by expression of the pan-autonomic homeobox gene *Phox2b* and are located, respectively, in the intermediate reticular formation of the medulla and around the motor nucleus of the trigeminal nerve. They are premotor to all jaw-opening and tongue muscles. Stimulation of either, in awake animals, opens the jaw, while IRt$^{Phox2b}$ alone also protracts the tongue. Moreover, stationary stimulation of IRt$^{Phox2b}$ entrains a rhythmic alternation of tongue protraction and retraction, synchronized with jaw opening and closing, that mimics lapping. Finally, fiber photometric recordings show that IRt$^{Phox2b}$ is active during volitional lapping. Our study identifies one of the subcortical nuclei underpinning a stereotyped feeding behavior.

[1] Institut de Biologie de l'ENS (IBENS), Inserm, CNRS, École normale supérieure, PSL Research University, Paris, France. [2] Max Delbrück Center for Molecular Medicine in the Helmholtz Association (MDC), and Neuroscience Research Center, Charité-Universitätsmedizin, Berlin, Germany. [3] Université Paris-Saclay, CNRS, Institut des Neurosciences NeuroPSI, Gif-sur-Yvette, France. [4] Institut Pasteur, INSERM, Institut de l'Audition, Paris, France. [5] Developmental Biology/ Signal Transduction, Max Delbrueck Center for Molecular Medicine, and Cluster of Excellence NeuroCure, Neuroscience Research Center, Charité-Universitätsmedizin, Berlin, Germany. [6] Faculty of Medicine, Health & Human Sciences, Macquarie University, Macquarie Park, NSW, Australia. [7] These authors contributed equally: Gilles Fortin and Jean-François Brunet. ✉email: jfbrunet@biologie.ens.fr

The hindbrain (medulla and pons) is a sensory and motor center for the head and the autonomic (or visceral) nervous system. Large areas therein defy conventional cytoarchitectonic description and are subsumed under the label "reticular formation"[1]. Over decades, the reticular formation has slowly emerged from "localizatory nihilism"[2], and regions defined by stereotaxy [e.g., ref. [3]], or cell groups defined by their projections [e.g., ref. [4]] have been implicated in a variety of roles: premotor neurons to orofacial or respiratory muscles[5, 6], and— underpinning the sophisticated residual behaviors observed in decerebrate animals[7]—rhythm and pattern generators for chewing, whisking, breathing, and sighing[3, 5, 8–11]. Licking is another rhythmic behavior for which a hindbrain rhythm generator is predicted[12] although the evidence is mostly extrapolated from chewing, the two behaviors possibly sharing some neuronal substrate[9].

However, the parsing of the reticular formation into genetically defined neuronal groups, endowed with specific connectivity and roles, has only begun[13–17] and lags behind other parts of the brain, such as the cortex or the spinal cord. Among the most specific genetic markers of neuronal classes are transcription factors, in particular, homeodomain proteins [e.g., refs. [18, 19]]. *Phox2b* is one such gene, which marks (and specifies) a limited set of neurons in the peripheral nervous system and the hindbrain, including the reticular formation. The expression landscape of *Phox2b* is strikingly unified by physiology: most *Phox2b* neurons studied to date, partake in the sensorimotor reflexes of the autonomic nervous system, that control bodily homeostasis[20]. An apparent exception is branchial motor neurons, that motorize the face and neck[1, 21] but their kinship to visceral circuits, aptly highlighted by their alternative name of "special visceral", is revealed by their exclusive ancestral functions in aquatic vertebrates, in feeding and breathing—thus visceral indeed. To this broadened picture of the visceral nervous system, in charge of vital functions and maintenance of the interior milieu, we now add two groups of *Phox2b* interneurons, located in the reticular formation of the hindbrain, that are premotor to orofacial muscles and can command licking or lapping, a rhythmic feeding behavior essential for the intake of liquids in many terrestrial vertebrates.

## Results

**The reticular formation harbors *Phox2b*$^+$ orofacial premotor neurons**. We visualized the total projections of *Phox2b* interneurons that are located in the reticular formation. The vast majority of these neurons are glutamatergic, thus express the glutamate vesicular transporter *Vglut2*, as shown by expression of the *Cre* and *Flpo*-dependent reporter *RC::Fela* in a *Phox2b::Flpo;Vglut2::Cre* background (Supplementary Fig. 1a). We used this neurotransmitter phenotype to implement an intersectional strategy that excludes the potentially confounding widespread projections of other *Phox2b*$^+$ neurons, in the locus coeruleus[22], which are noradrenergic. We designed an intersectional allele (*Rosa$^{FRTtomato-loxSypGFP}$* or *Rosa$^{FTLG}$*) (Fig. 1a) which expresses one of two fluorophores, exclusively: the action of flippase (*Flpo*) will trigger cytoplasmic expression of *tdTomato* (*tdT*), while additional action of *Cre* recombinase, will extinguish *tdT* in the cell soma and switch on instead a fusion of synaptophysin with GFP (*Syp-GFP*) transported to presynaptic sites[23]. When *Flpo* was driven by the *Phox2b* promoter, and *Cre* by the *Vglut2* promoter, i.e., in *Phox2b::Flpo;vGlut2::Cre;Rosa$^{FTLG}$* pups, at P4 tdT was expressed, as expected, in the soma of the singly recombined motoneurons (which are *Phox2b*$^+$, but not glutamatergic), but lost from the doubly recombined interneurons (which are *Phox2b*$^+$ and glutamatergic) (Supplementary Fig. 1b). The latter, in turn,

had switched on *Syp-GFP* in their synaptic boutons, which covered remarkably discrete structures of the hindbrain (Supplementary Fig. 1b and Fig. 1b), among which motor nuclei (whose function will be discussed later) featured prominently: (i) most branchiomotor (*Phox2b*$^+$) nuclei—the trigeminal motor nucleus (Mo5) and its accessory nucleus (Acc5), the facial nucleus (Mo7) (albeit only its intermediate lobe) and its accessory nucleus (Acc7), the nucleus ambiguus (MoA); (ii) two somatic (*Phox2b*$^-$) motor nuclei: the hypoglossal nucleus (Mo12), and a nucleus in the medial ventral horn, at the spinal-medullary junction, which innervates the infrahyoid muscles[24] (and Supplementary Fig. 1c), and that we call MoC (to denote its projection through the upper Cervical nerves)[24]. Other cranial motor nuclei were free of input from *Phox2b*$^+$/*vGlut2*$^+$ interneurons: those for extrinsic muscles of the eye (oculomotor (Mo3) and trochlear (Mo4)), and for the spinal accessory nucleus (Mo11), which innervates the sterno-cleidomastoid and trapezius muscles (Supplementary Fig. 1d). The abducens nucleus (Mo6) however, did receive boutons (Supplementary Fig. 1d). Thus, somewhere in the reticular formation, are *Phox2b*$^+$ orofacial premotor neurons, which we then sought to locate.

To locate *Phox2b*$^+$ orofacial premotor neurons, we used retrograde transsynaptic viral tracing from oromotor muscles. We injected a G-defective rabies virus variant encoding the fluorophore *m-Cherry*[25] together with a helper virus encoding G and the fluorophore YFP (*HSV-YFP-G*) in the posterior belly of the digastric muscle (Fig. 1c) (a jaw-abductor), known to be innervated by Acc7[26, 27]. Predictably, the only seed neurons (i.e., that co-express the rabies virus-encoded mCherry and the helper virus-encoded YFP) were found in Acc7 (right panel in Fig. 1c). Premotor neurons, presynaptic to the seed motoneurons (i.e., that express only the rabies virus-encoded mCherry) and which, in addition, were *Phox2b*+, were found at two sites only: (i) the intermediate reticular formation (IRt) (Fig. 1d) and (ii) "regio h", arranged in "shell form" around Mo5[28], more commonly called the peritrigeminal region (Peri5)[29] (Fig. 1e). We found the same pattern of *Phox2b*$^+$ premotor neurons for the geniohyoid muscle (a hyoid protractor and jaw-abductor) (Supplementary Fig. 2a), innervated by the accessory compartment of Mo12 (Acc12)[24]; and we found a subset of this pattern for the genioglossus (a tongue protractor and/or jaw-abductor) (Supplementary Fig. 2b) and for the intrinsic muscles of the tongue (Supplementary Fig. 2c) (both innervated by Mo12), whereby *Phox2b*$^+$ premotor neurons were restricted to the IRt. On the other hand, the masseter (the main jaw-closing muscle) and the thyroarytenoid (that motorizes the vocal cords) had totally distinct premotor landscapes (Supplementary Fig. 2d, e)[30, 31].

We next sought to characterize genetically and developmentally the *Phox2b*$^+$ orofacial premotor neurons located in Peri5 and IRt.

**Transcriptional signature and developmental origin of Peri5$^{Atoh1}$ and IRt$^{Phox2b}$**. The *Phox2b*$^+$ premotor nucleus that occupies Peri5, we shall call Peri5$^{Phox2b}$ (Fig. 2a, b). Because it surrounds, shell-like, a nucleus with a history of *Phox2b* expression(Mo5 + Acc5)it cannot be selectively accessed with *Phox2b*-based tools, even refined by stereotaxy. We thus restricted our study to a distinct subnucleus of Peri5$^{Phox2b}$, which unlike the rest of the nucleus co-expresses *Phox2b* with another transcription factor, *Atoh1*[32] and that we shall call Peri5$^{Atoh1}$ (Fig. 2b–d). Peri5$^{Atoh1}$ is made of $2052 \pm 184$ cells ($n = 4$) at late gestation (E18.5), is premotor to the posterior digastric (Supplementary Fig. 2f), and can be selectively targeted in an intersectional *Phox2b::Flpo;Atoh1::Cre* background[16, 33] (Fig. 2e). Peri5$^{Atoh1}$ cells express *Lbx1* (Fig. 2f), thus originate from the dB progenitor domain[34]. More precisely

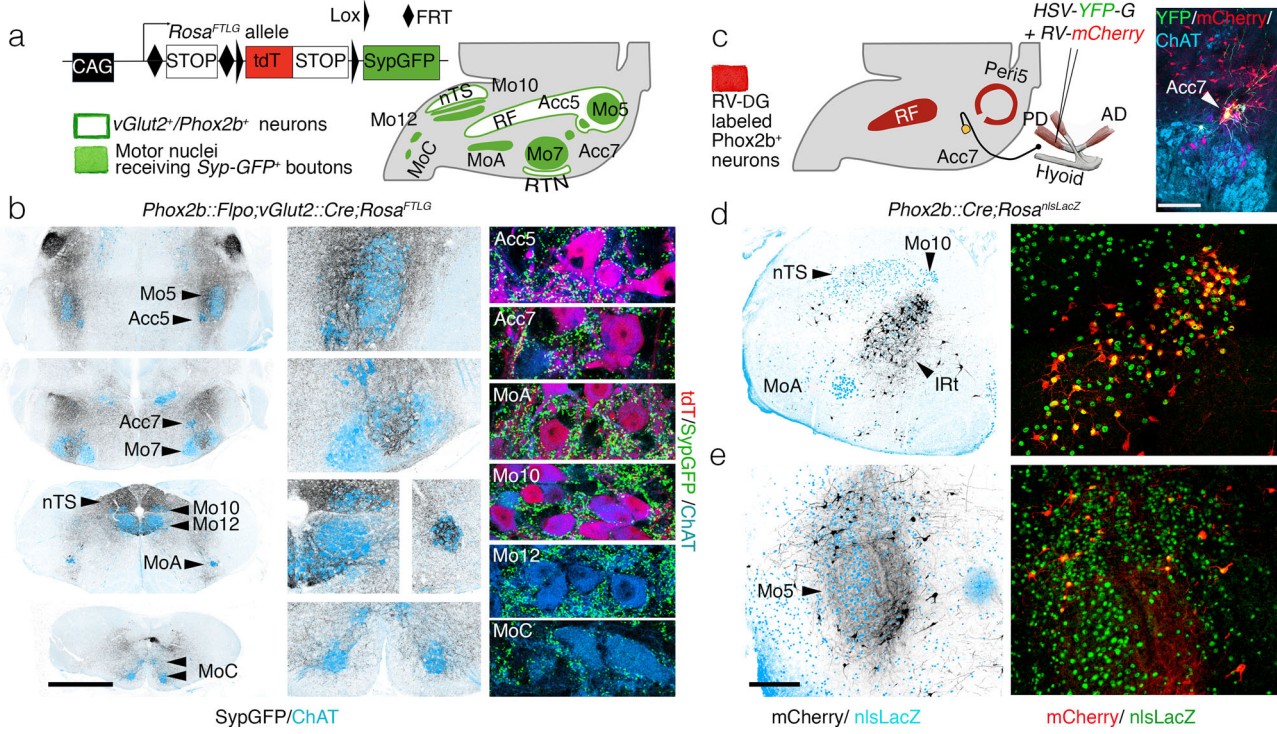

**Fig. 1 Premotor status of reticular formation _Phox2b_+interneurons. a** _Rosa^FTLG_ allele used for intersectional transgenic labeling of boutons from _vGlut2/Phox2b_ interneurons (left) and schematic of the results (right). **b** Coronal sections through the hindbrain of a _Phox2b::Flpo; vGlut2::Cre;Rosa^FTLG_ mouse at P4, showing synaptic boutons (black) from _vGlut2/Phox2b_ interneurons in relation to motor nuclei (ChAT+, blue) at low (left), and higher (middle) magnifications, and close-ups of boutons (green) on motoneurons (right), which are either _Phox2b_+ (purple) or _Phox2b_− (blue). **c** (left) Strategy for mono-synaptically restricted transsynaptic labeling of premotor neurons from the posterior digastric muscle (PD) in a _Phox2b::Cre;Rosa^nlsLacZ_ mouse, with G-deleted rabies virus (RV) encoding mCherry and complemented by a G-encoding helper HSV virus (_HSV-YFP-G_), and summary of the results. (right panel) The only seed neurons are Acc7 motoneurons, double-labeled by the _HSV-G_ and _RV-mCherry_ viruses. **d**, **e** Coronal sections through the hindbrain at P8 showing labeled premotor neurons (black on the left panels) in the IRt (**d**) and Peri5 (**e**), which for the most part (72.7% ± 3.5 SEM, _n_ = 4 animals) express _Phox2b_ (right panels). AD anterior digastric, IRt intermediate reticular formation, nTS nucleus of the solitary tract, PD posterior digastric, Peri5 peritrigeminal area, RF reticular formation, RTN retrotrapezoid nucleus. Scale bars, **b** 1 mm for the left column, **c** 250 μm, **d**, **e** 500 μm.

they belong to its dB2 derivatives, at the leading edge of whose migration stream they become detectable at E11.5, near the incipient Mo5 (Fig. 2g).

The _Phox2b_+ premotor nucleus that occupies IRt, we shall call IRt^Phox2b (Fig. 2a). It shares with the nearby nTS the _Phox2b_+/_Tlx3_+/_Lmx1b_+ signature and an origin in _Olig3_+ progenitors (i.e., the pA3 progenitor domain[35]) (Fig. 2a, h). It is distinguished, however, by the expression of the transcriptional cofactor _Cited1_ (Fig. 2i). IRt^Phox2b segregates topographically from nTS at E13.5 (Fig. 2i) from which it can thus be told apart by stereotaxy. The border between the two nuclei is marked by the intramedullary root of Mo10 (Fig. 2j). Unlike nTS, IRt^Phox2b does not receive any input from the tractus solitarius (Fig. 2k). Also unlike the nTS, IRt^Phox2b neurons are intermingled with glutamatergic neurons of other types (_Phox2b_-negative) (Supplementary Fig. 3). Thus, IRt^Phox2b and nTS are two structures related by lineage, which acquire distinct molecular, topological, and hodological identities.

**Peri5^Atoh1 and IRt^Phox2b target jaw opening and tongue muscles**. We confirmed the premotor status of Peri5^Atoh1 and IRt^Phox2b in adult animals by anterograde tracing with viral and transgenic tools (Fig. 3). For Peri5^Atoh1, we used the _Rosa^FTLG_ allele recombined by _Phox2b::Flpo_[33] and _Atoh1::Cre_[16] (Fig. 3a). The GFP+ boutons covered Acc5, intermediate Mo7, Acc7, Mo10, Mo12, and MoC (Fig. 3a–f). In Mo12, the rostro-ventral compartment was excluded (Fig. 3d, e). Because the retrotrapezoid nucleus (RTN) is also _Atoh1_+/_Phox2b_+ [16], thus could

confound this pattern, we confirmed the projections of Peri5^Atoh1 by anterograde tracing with a _Cre_-dependent adeno-associated virus (AAV) expressing _mGFP_ and _Syp-mRuby_[36] injected in Mo5 of a mouse harboring both, _Phox2b-Flpo_ and an _Atoh1-Cre_ that is dependent on _Flpo_ (_Atoh1::FRTCre_)[16] (Supplementary Fig. 4a, b). Using the same vector, this time stereotaxically injected in IRt^Phox2b of a _Phox2b::Cre_ mouse, we found the projections from IRt^Phox2b in the same motor nuclei as those from Peri5^Atoh1 (Fig. 3g–l)— with the sole difference that in Mo12, the ventral compartment was targeted, rather than the dorsal one (compare Fig. 3j, k with Fig. 3d, e).

To map putative collaterals of _Phox2b_+ premotor neurons, we performed a retrograde transsynaptic tracing experiment from the posterior digastric in a genetic background that, in addition, labels the boutons of all _Phox2b_+ neurons with GFP (_Phox2b::Cre;Rosa::Syp-GFP_) (Fig. 3m). Double-labeled terminals (_m-Cherry_+; _Syp-GFP_+)—thus, sent by neurons that are both, _Phox2b_+ and premotor to the posterior digastric—were found, in addition to Acc7 (the motor nucleus of the injected muscle), in Acc5, intermediate Mo7, Mo12, and MoC (Fig. 3n–q). Thus, _Phox2b_+ orofacial premotor neurons to Acc7 are collateralized in a way that hardwires Acc5, intermediate Mo7, Acc7, Mo12, and MoC to activate their target muscles together.

The combined action of head motor nuclei innervated by Peri5^Atoh1 and IRt^Phox2b should mobilize the jaw, lower lip and tongue: Acc5 and Acc7 innervate the four suprahyoid muscles[37–39], which depress the jaw via the hyoid apparatus. Intermediate Mo7 innervates the _platysma_[39], probably a jaw

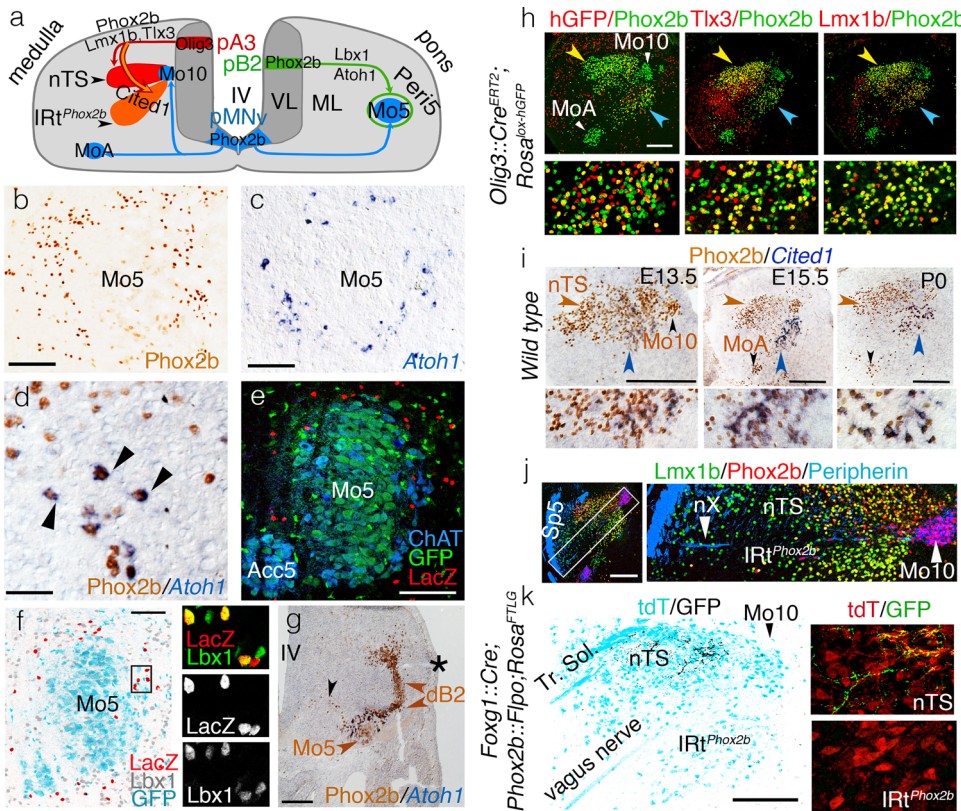

**Fig. 2 Ontogenetic definition of IRt^{Phox2b} and Peri5^{Atoh1}. a** Two schematic hemisections of the embryonic medulla (left) or pons (right), showing the origin of branchiomotor nuclei (Mo5, MoA, and Mo10), Peri5^{Phox2b} and IRt^{Phox2b} in progenitor (p) domains of the ventricular layer (VL), their settling sites in the mantle layer (ML), and their transcriptional codes. **b–d** Coronal sections through the pons at E18.5, showing Peri5^{Phox2b} (**b**) or Peri5^{Atoh1} (**c**, **d**) labeled with the indicated antibody or probe. Peri5^{Atoh1} cells co-express *Phox2b* and *Atoh1* (arrowheads in **d**). **e** Coronal sections through Mo5 in a *Phox2b::Flpo;Atoh1::Cre;Fela* mouse at P0, showing the doubly recombined (nlsLacZ^+) cells of Peri5^{Atoh1} (red). **f** Coronal section through Mo5 in a *Phox2b::Flpo;Atoh1::Cre;Fela* mouse, where *Phox2b^+* motoneurons are GFP^+ (cyan) and *Phox2b^+/Atoh1^+* neurons are nlsLacZ^+ (red), counterstained for Lbx1 (gray at low magnification, green in the close-ups). **g** Coronal section through the pons at E11.5 showing the migrating *Phox2b^+* Mo5 and dB2 precursors (black and brown arrowheads, respectively) and, at their meeting point, Peri5^{Atoh1} cells that have switched on *Atoh1*. Asterisk: lateral recess of the IVth ventricle (IV). **h** Coronal sections through nTS (yellow arrowhead) and IRt^{Phox2b} (blue arrowhead) at E18.5, at low magnification (upper) or at high magnification for the IRt (lower), stained with the indicated antibodies. A history of *Olig3* expression is revealed by recombination of the histone-GFP (hGFP) reporter in the *Olig3::Cre^{ERT2}* background (left). Mosaicism is likely due to incomplete induction of Cre. Virtually all cells of IRt^{Phox2b} (98% ± 0.2 SEM, *n* = 3 animals) co-expressed *Lmx1b* with *Phox2b*. **i** Coronal sections through nTS (brown arrowhead) and IRt^{Phox2b} (blue arrowhead) at indicated stages at low magnification (upper) and high magnification for the IRt (lower), immunostained for Phox2b and in situ hybridized for *Cited1*. **j** Coronal section at E15.5 showing that nTS and IRt^{Phox2b} are separated by the medullary root of the vagus nerve (nX). Sp5 spinal trigeminal tract. **k** Coronal section through the nTS and IRt^{Phox2b} of an adult, showing the central boutons of epibranchial ganglia (that express *Foxg1*[72] and are labeled by SypGFP in a *Foxg1^{iresCre};Phox2b::Flpo;Rosa^{FTLG}* background) in the nTS, but not IRt^{Phox2b} (left). Magnified details (right). Scale bars, **b**, **c** 125 µm, **d** 50 µm, **e**, **f** 100 µm, **g**, **h**, **j**, **k** 200 µm, and **i**, 250 µm.

depressor[40], and a *mentalis*[39], which, together with the *platysma*, pulls down the lower lip. Ventral Mo12, targeted by IRt^{Phox2b}, innervates tongue protractors[41], while dorsal Mo12, targeted by Peri5^{Atoh1}, innervates tongue retractors[42]. Finally, MoC innervates the infrahyoid muscles, classically viewed as stabilizers of the hyoid during jaw lowering, but which probably collaborate with the suprahyoids in a more complex fashion[43]. Thus, Peri5^{Atoh1} and IRt^{Phox2b} appear connected so as to, collectively, lower the jaw, while retracting or protracting the tongue, respectively.

In addition, anterograde tracing from IRt^{Phox2b} in a *Phox2b::Cre* background and from Peri5^{Atoh1} in a *Phox2b::Flpo;Atoh1::Cre* background revealed, respectively, massive projections of IRt^{Phox2b} to the peri5 region (Fig. 3h) and of Peri5^{Atoh1} to the IRt region. (Supplementary Fig. 4c). We could not assess the precise cellular targets of IRt^{Phox2b}, but those of Peri5^{Atoh1} included IRt^{Phox2b} (Supplementary Fig. 4d, inset), suggesting reciprocal connections of the two nuclei.

**Peri5^{Atoh1} and IRt^{Phox2b} can trigger tongue and jaw movements.** We optogenetically stimulated IRt^{Phox2b} or Peri5^{Atoh1} in head-fixed awake animals. To do so, we injected a *Cre*-dependent AAV that directs expression of the soma-targeted excitatory opsin stCoChR, either in IRt^{Phox2b} of *Phox2b::Cre* mice (Fig. 4a) or in Peri5^{Atoh1} of *Phox2b::Flpo;Atoh1^{FRTCre}* mice (Fig. 4b). Single light pulses (100 ms) on IRt^{Phox2b} evoked a wide opening of the mouth accompanied by tongue protraction, which terminated upon cessation of the pulse (Fig. 4a), while the same stimulus applied to Peri5^{Atoh1} triggered only mouth opening, of smaller amplitude (Fig. 4b). Thus, both nuclei can open the mouth, in agreement with their projections on the motoneurons for the suprahyoid and infrahyoid muscles (Fig. 1b and Supplementary Figs. 2a, 3), while IRt^{Phox2b} but not Peri5^{Atoh1} can protract the tongue, in line with the targeting of hypoglossal motoneurons for tongue protractors by the former and tongue retractors by the latter (Fig. 3d, e, j, k). Delivering the stimulus at 4, 5, or 7 Hz led to a faithful repetition of the movement (Supplementary Fig. 5a)

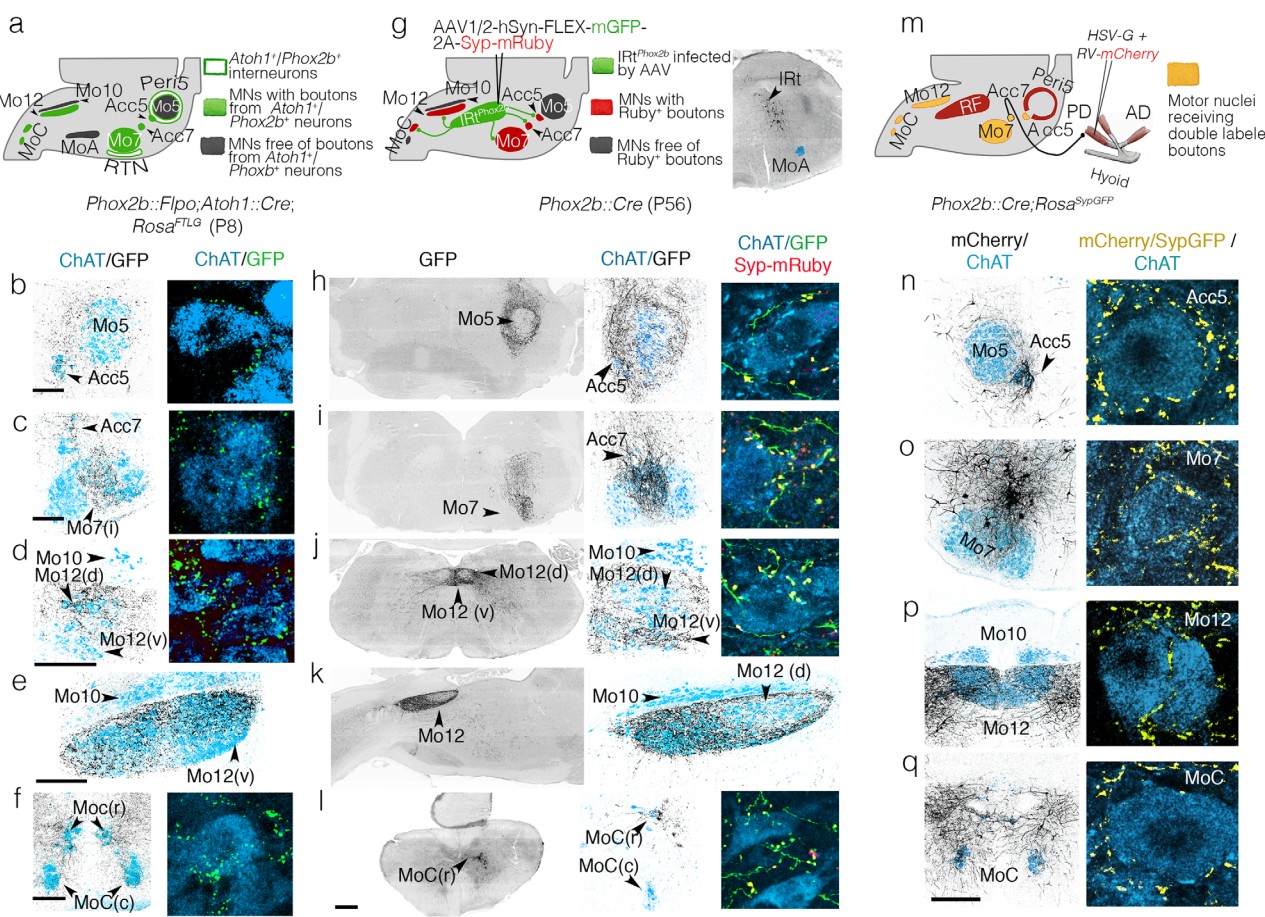

**Fig. 3 Projections of IRt$^{Phox2b}$ and Peri5$^{Atoh1}$ on hindbrain motoneurons. a** Strategy for the transgenic labeling of projections from Peri5$^{Atoh1}$ (and RTN) and summary of the results; **b–f** Coronal (**b–d**, **f**) or parasagittal (**e**) sections through a P8 hindbrain showing *GFP*-labeled boutons (black) on motoneurons (blue) at medium (left) and high (right) magnification. **g** Strategy for the viral tracing of projections from IRt$^{Phox2b}$ and summary of the results (left), and mGFP-labeled infected cells of IRt$^{Phox2b}$ (right); **h–l** Coronal (**h–j**, **l**) or parasagittal (**k**) sections through a P56 hindbrain showing the *GFP*-labeled fibers (black) of IRt$^{Phox2b}$ neurons at low (left) and medium (middle) magnifications, and in extreme close-ups (right), together with Syp-mRuby labeled boutons (yellow) on motoneurons (blue). **m** schematic for retrograde tracing of premotor neurons for the right posterior digastric muscle, in a *Phox2b::Cre;Rosa$^{SypGFP}$*, and summary of the results. **n–q** (left) Coronal sections through the hindbrain at P8 showing the *mCherry$^+$* projections (black) of premotor neurons on the motor nuclei (ChAT$^+$, blue); (right) close-ups on motoneurons receiving double-labeled *Syp-GFP/mCherry* boutons (yellow). Scale bars, **b–f** 200 μm for the left column, **h–l** 500 μm for the left column, **n–q** 200 μm for the left column.

showing that IRt$^{Phox2b}$ can operate in this frequency range. As expected from the premotor status of IRt$^{Phox2b}$, lengthening the light pulse on IRt$^{Phox2b}$ to 200 ms analogically prolonged the mouth opening and tongue protraction (Fig. 4c). Unexpectedly, however, further lengthening led to the termination of the initial movement and its rhythmic repetition at around 7 Hz (Fig. 4c, Supplementary Fig. 5b, and Supplementary Movie 1), a frequency similar to that of naturally occurring licking (Supplementary Fig. 5c)[44]. Conversely, prolonged illumination of Peri5$^{Atoh1}$ only prolonged the initial mouth opening (Fig. 4d, Supplementary Fig. 5d, and Supplementary Movie 2). Thus, a contrast between the actions of photo-stimulated Peri5$^{Atoh1}$ and IRt$^{Phox2b}$ lies in the ability of the latter to translate stationary excitation into a rhythmic series of oromotor movements, akin to naturally occurring licking[44].

**IRt$^{Phox2b}$ is active during volitional licking.** We then tested whether IRt$^{Phox2b}$ is active during spontaneous fluid ingestion. We recorded the bulk fluorescence[45] of IRt$^{Phox2b}$ in head-fixed *Phox2b::Cre* mice, injected in IRt$^{Phox2b}$ with a *Cre*-dependent AAV encoding the calcium indicator jGCamp7s[46] and implanted

with an optical cannula (Fig. 4e). During freely initiated bouts of licking from a water-spout, we observed a systematic increase in fluorescence of IRt$^{Phox2b}$ immediately upon deflection of the jaw that preceded individual licks or bouts of lapping (Fig. 4f, g, Supplementary Fig. 5e, and Supplementary Movie 3). Thus, IRt$^{Phox2b}$ neurons, capable of triggering a licking behavior with physiological frequency, are active during such spontaneous behavior. Importantly, IRt$^{Phox2b}$ encompasses the location of many neurons previously identified as rhythmically active during licking[9]. Stationary optogenetic stimulation of this nucleus might emulate the effect of sustained drive from the licking area of the oromotor cortex[47–50].

**Inputs to IRt$^{Phox2b}$.** Although decerebrated mammals can display reflexive licking[7, 51], volitional or self-initiated licking requires higher brain centers. To explore the substratum for this requirement, we traced the inputs to IRt$^{Phox2b}$ by co-injecting it with a pseudotyped G-defective rabies virus variant encoding *m-Cherry* and a helper virus that depends on *Cre*, in a *Phox2b::Cre* background (Fig. 5a). The vast majority of inputs (about 90%) were in the brainstem (Fig. 5b), which could explain

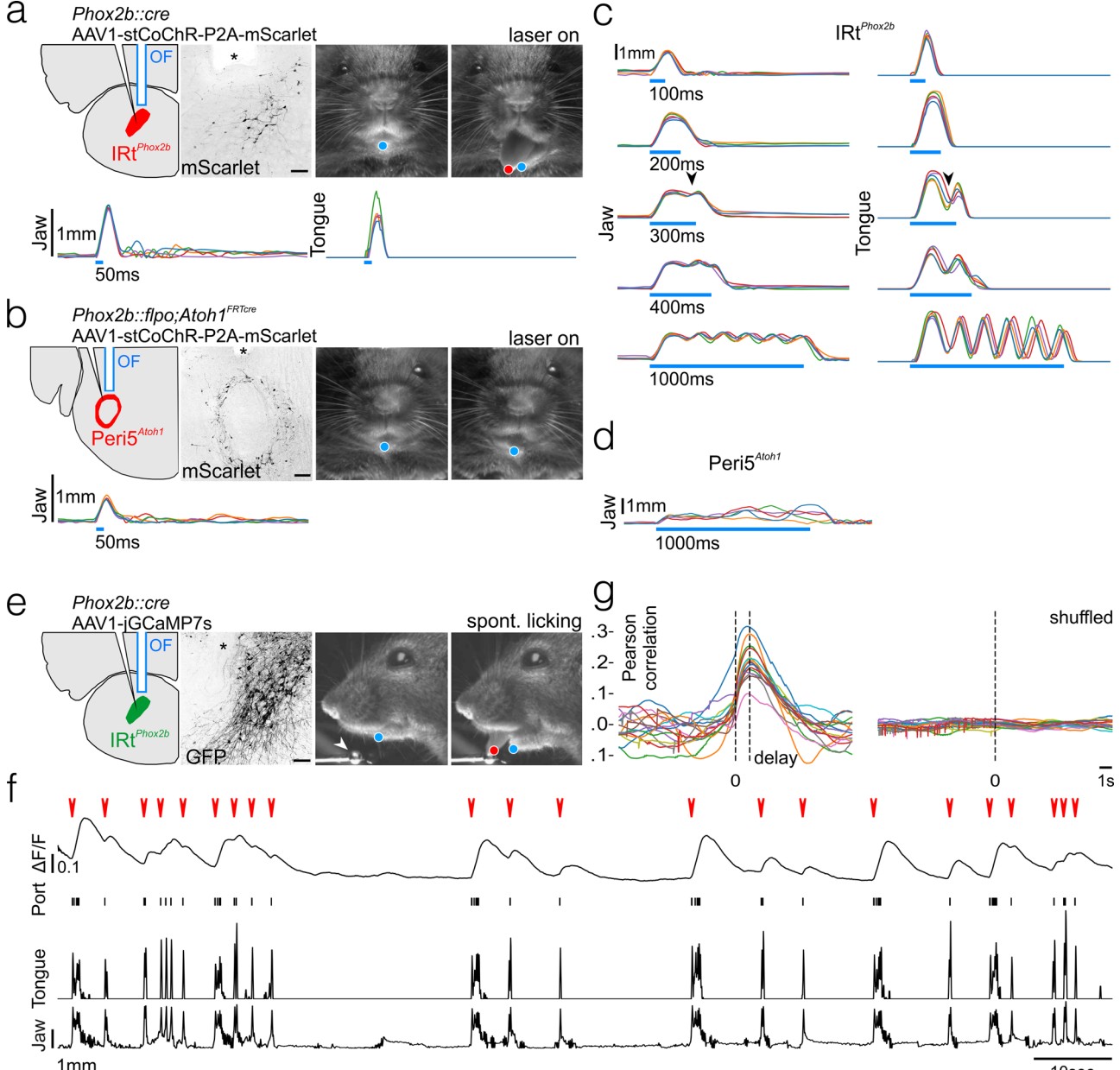

**Fig. 4 Orofacial movements triggered by IRt^Phox2b and Peri5^Atoh1 and activity of IRt^Phox2b during voluntary licking. a** (Upper left) Schematic of the viral injection and fiber-optic implantation for stimulation of IRt^Phox2b, and transverse section through the hindbrain showing transduced IRt^Phox2b neurons and position of optical fiber (OF, asterisk); scale bar 100 μm. (Upper right) Example frames of the mouse face before and during stimulation including DeepLabCut tracked position of the jaw (blue) and tongue (red). (Lower) Individual traces of tracked jaw and tongue position on the Y-axis upon 50 ms stimulation (five trials). **b** (Upper left) Schematic of the viral injection and fiber-optic implantation for stimulation of Peri5^Atoh1 and transverse section through the hindbrain showing transduced Peri5^Atoh1 neurons and position of optical fiber (asterisk); scale bar 200 μm. (Upper right) Example frames of the mouse face before and during stimulation including DeepLabCut tracked position of jaw (blue). (Lower) Individual traces (five trials) of tracked jaw position on the Y-axis upon 50 ms stimulation. **c** Individual traces (five trials) of the tracked jaw (left) and tongue (right) position on the Y-axis upon stimulation of IRt^Phox2b of increasing length. A repetitive movement is triggered by stimulation beyond 300 ms (arrowhead). **d** Individual traces (five trials) of tracked jaw position on the Y-axis upon a 1000 ms stimulation of Peri5^Atoh1. The jaw remains open and quivers non-rhythmically during the stimulus. **e** (Left) Schematic of viral injection and optical fiber implantation for observation of IRt^Phox2b activity, and transverse section through the hindbrain showing transduced IRt^Phox2b neurons and position of optical fiber (asterisk); scale bar 100 μm. (Right) Example frames of the mouse face before and during a bout of licking from a lick port (arrowhead), during a photometry recording, including DeepLabCut tracked position of the jaw (blue) and tongue (red). **f** Example trace of change in bulk fluorescence of IRt^Phox2b during a recording session (~2 min) of unitary licking events and licking bouts (red arrowheads), contact events with the lick port, and movements of the tongue and the jaw on the Y-axis. **g** (left) Superimposed correlation curves between licking activity and calcium activity (each curve corresponding to one of 15 recording sessions, each 1–5 min, in one mouse) which peaked at 1.2 s after lick port contact; (right) no peak was observed after shuffling the data.

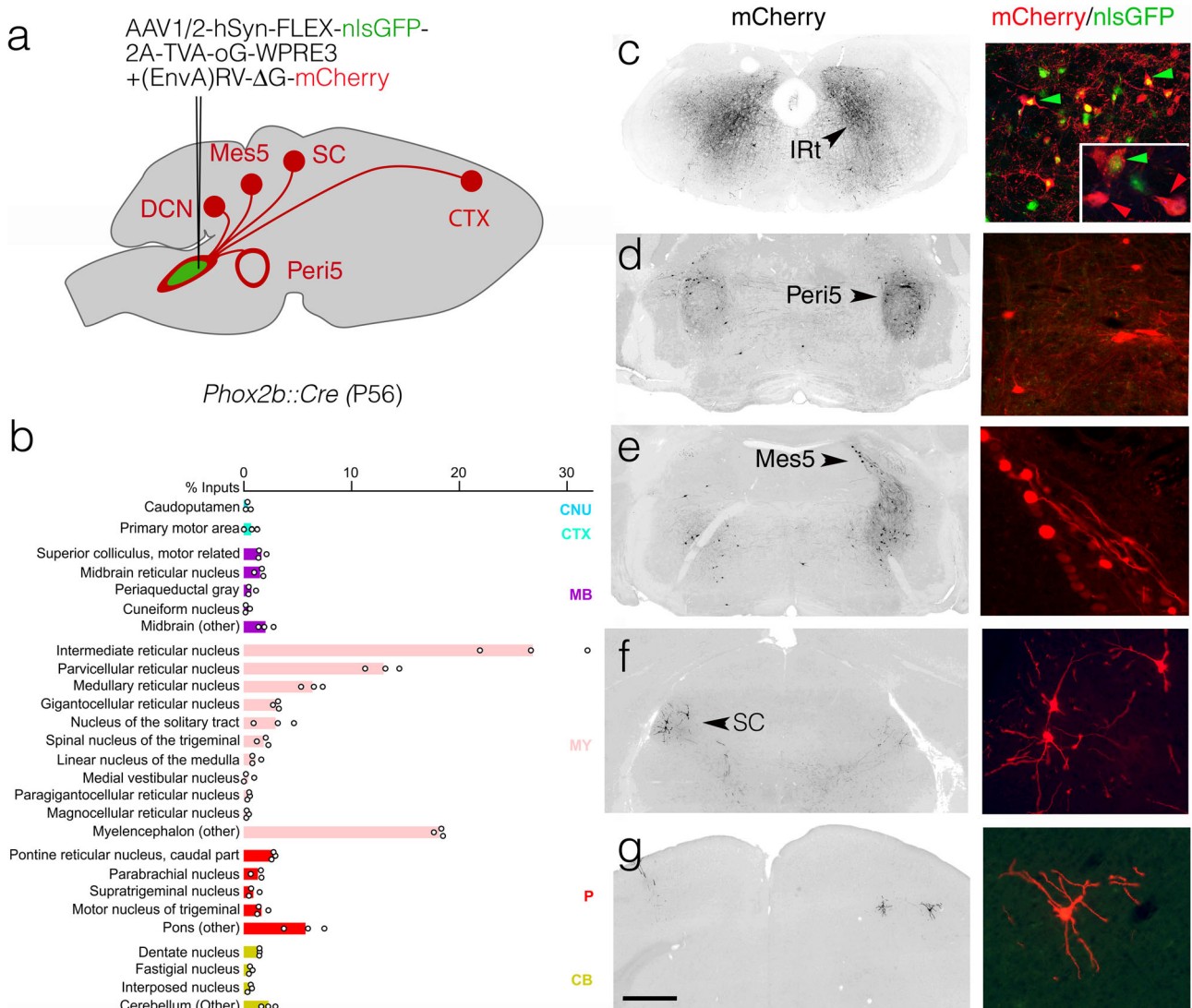

**Fig. 5 Inputs to IRt$^{Phox2b}$. a** Strategy for the retrograde transsynaptic labeling of input neurons to IRt$^{Phox2b}$ with exemplar sites of input. **b** Bar graph of the relative percentage of monosynaptic input neurons labeled from IRt$^{Phox2b}$ starter neurons, displayed per brain region as defined in the Allen Brain Atlas ($n = 4063$ input cells and $n = 598$ starter cells, from $n = 3$ animals; individual values as circles; seeding efficiency $= 7.4$ inputs/starters $\pm 1.8$ SEM). Rabies labeled input neurons were largely ($74.5 \pm 1.1\%$ SEM) restricted to the medulla (pink, MY) and exhibited a slight but consistent ipsilateral bias ($55.6 \pm 3.0\%$ SEM). Major sources of these medullary inputs were the intermediate, gigantocellular, and parvocellular reticular nuclei. Inputs from the cortex, midbrain, and pons represented a minority of rabies-labeled neurons ($1.6 \pm 0.3\%$ SEM, $6.4 \pm 0.5\%$ SEM, and $12.3 \pm 1.1\%$ SEM respectively). **c–g** Images at low magnification (left) and high magnification (right) of monosynaptic input neurons in the IRt, Peri5, mesencephalic nucleus of the trigeminal nerve, contralateral superior colliculus, and motor cortex. Green arrowheads: seed neurons; red arrowheads: ($n$-1) IRt neurons. CB cerebellum, CNU caudoputamen, CTX cortex, DCN deep cerebellar nucleus, MB midbrain, Mes5 mesencephalic nucleus of the trigeminal nerve, MY myelencephalon, P pons, SC superior colliculus. Scale bar **c–g** 1 mm.

the largely intact reflexive behavior of decerebrated animals. Among these regional inputs, many were found in IRt itself, including contralaterally (Fig. 5c)—suggesting local inter-connectivity of IRt neurons, possibly related to rhythmogenesis, through recurrent synaptic connections, as hypothesized for other rhythm generating structures[52]. Other regional inputs came from the peri5 region (Fig. 5d)—likely including Peri5$^{Atoh1}$ that we had traced anterogradely to IRt$^{Phox2b}$ (Supplementary Fig. 4d)—the mesencephalic nucleus of the trigeminal nerve (Mes5) (Fig. 5e)—which harbors proprioceptors for the teeth and masseter, potentially allowing for a cross talk between jaw position and tongue movement[53]—and the superior colliculi (Fig. 5f)—whose inhibition disrupts self-initiated licking[54]. Finally, we found input from the cortex (Fig. 5g), where a subclass of pyramidal tract

neurons are known to directly target orofacial promotor neurons[49].

## Discussion

Our study uncovers two genetically coded neuronal groups in the reticular formation, involved in orofacial movements. They are premotor to orofacial muscles and collateralized, thus in a position to coordinate the contraction of a precise set of muscles to the exclusion of others, a property previously highlighted in studies of orofacial premotor neurons (refs. [5], [6] and references therein). As such, they represent an essential hierarchical level in the orchestration of complex oropharyngeal behaviors. In addition, one of them, IRt$^{Phox2b}$, translates a tonic stimulation into a

rhythmic behavior. The most parsimonious interpretation of IRt$^{Phox2b}$ is that its neurons are bifunctional: premotor through their collateralized inputs on motor nuclei, and rhythm generators, corresponding to the hypothetical licking CPG[12] or at least an element thereof, in the precise region where many lick-rhythmic neurons were previously recorded (refs. [8, 9] for reviews). It is of note that another nearby Phox2b$^+$ nucleus, the RTN, has intrinsic rhythmic properties, in that case related to breathing, in the neonate[15, 55]. At this stage, though, we cannot exclude that IRt-$^{Phox2b}$ contains two subtypes of neurons, one premotor and the other pre-premotor, and that it is the latter which, upon photo-stimulation, triggers rhythmic repetition; in other words, that IRt$^{Phox2b}$ encompasses a two (or more)-level architecture, akin to models proposed for other motor behaviors[56–59]. This possibility is made less likely by the apparent genetic homogeneity of IRt-$^{Phox2b}$, whose neurons all co-express the transcriptional signature Phox2b/Cited1. Finally, the possibility that the rhythm would be generated by neurons elsewhere in the brainstem (recruited by IRt$^{Phox2b}$ and feeding back on it) is constrained by the limited output of IRt$^{Phox2b}$: to motor nuclei and the peri5 region.

In addition to rhythmic tongue protrusion and jaw opening, the entrainment of a full licking cycle requires the delayed activation of antagonistic muscles (as in several "burst generator" models of the locomotor CPG, e.g. ref. [58]). One substrate for such rhythmic alternation might comprise the reciprocal projections of IRt$^{Phox2b}$ and Peri5$^{Atoh1}$ (Fig. 3h and Fig. S3C, D), the former targeting tongue protractors and the latter tongue retractors.

From a developmental and evolutionary perspective, it is striking that IRt$^{Phox2b}$ and Peri5$^{Atoh1}$ express the pan-autonomic transcriptional determinant Phox2b, as do several of their motoneuronal targets. Thus, the evolutionarily conserved[60] selectivity of Phox2b for neurons involved in homeostasis, extends beyond the reflex control of the viscera, including all sensory-motor loops involved in digestion[20, 61], to the executive control of ingestion, through the Phox2b$^+$ premotor/motor arm that mobilizes visceral-arch derived muscles (Fig. 1 and Supplementary Fig. 1). The remarkable genetic monotony of these circuits breaks down at the level of the somatic (Phox2b$^-$) lingual and hypobranchials motoneurons. Such exceptions are to be expected in the head where the visceral and somatic bodies of the vertebrate animal, sensu Romer[62], must meet and cooperate, at the border of the external world and interior milieu. Indeed, feeding can be construed as a sequence of somatic (i.e., external or relational) and visceral (i.e., internal or homeostatic) actions: to take in a substrate from the environment by biting or licking/lapping up, then to incorporate it in the interior milieu by chewing and swallowing. In these actions, the hyoid bone act as a weld between visceral and somatic muscles of the head: respectively the suprahyoids, derived from visceral arch mesoderm and innervated by branchiomotor (Phox2b$^+$) motoneurons; and the hypobranchials (infrahyoid and lingual) derived from somites and innervated by somatic (Phox2b$^-$) motoneurons. The hyoid bone, branchiomeric muscles, branchiomotor neurons, and premotor centers Peri5$^{Atoh1}$ and IRt$^{Phox2b}$, all affiliated to the visceral body—muscles and bones through their origin in branchial arch mesoderm or neural crest, neurons through their expression of Phox2b—are likely the ancestral agents of feeding behaviors in vertebrates. At the advent of predatory and terrestrial lifestyles, the Phox2b$^+$ premotor centers must have recruited elements of the somatic body: the infrahyoid and lingual motoneurons, and their muscle targets, migrated into the head[63].

## Methods

**Mouse lines**. The following transgenic mouse lines were used: Phox2b::Flpo;[33] Phox2b::Cre[64], vGlut2::Cre[65], Atoh1::Cre[16], Atoh1::FRTCre[16], Olig3::Cre$^{ERT2}$ [35], Foxg1$^{iresCre}$[66], RC::FELA[67] Tau::Syp-GFP[68], Rosa::nlsLacZ (also known as

Tau$^{mGFP}$)[69], and Ai9[70]. For behavioral experiments, all mice were produced in a B6D2 background.

The Rosa$^{FTLG}$ mutant mouse line was established at the Institut Clinique de la Souris (Phenomin-ICS), Illkirch, France). The targeting vector was constructed as follows. A PCR fragment containing the rat synaptophysin cDNA fused to GFP was cloned by SLIC cloning with a 346 bp double-stranded synthetic HSV TK pA followed by a 29 bp homology for the 5′ extremity of the 3′ Rosa homology arm plus a NsiI site, in an ICS proprietary vector containing a floxed NeoR-STOP cassette. In the second cloning step, the NeoR cassette was removed by BamHI and SpeI restriction digests and replaced by SLIC cloning with the cDNA of tdTomato. The third cloning step introduced 5′ of the floxed tdTomato-STOP cassette, a DNA fragment containing a NsiI site followed by a 29 bp homology for the 3′ of the pCAG, followed by an MCS. The fourth step was the cloning of an FRT-surrounded NeoR-STOP cassette previously excised from an ICS proprietary vector in the SmaI site of the restriction site introduced in the MCS cassette. Finally, a fifth cloning step comprised the excision of a 7.8 kb fragment containing the whole FRT-NeoR-STOP-FRT LoxP-TdTomato-STOP-LoxP Syn-YFP cassette by a NsiI digest and its subcloning via SLIC cloning in an ICS proprietary vector containing a pCAG (Chicken b-actin promoter preceded by a CMV enhancer) and both 5′ and 3′ Rosa homology arms. The linearized construct was electroporated in C57BL/6 N mouse embryonic stem (ES) cells (ICS proprietary line). After G418 selection, targeted clones were identified by long-range PCR and further confirmed by Southern blot with an internal (Neo) probe and a 5′ external probe. One positive ES cell clone was validated by karyotype spreading and microinjected into BALB/c blastocysts. The resulting male chimeras were bred with wild-type C57BL/6 N females. Germline transmission was achieved in the first litter.

The sequence of all primers for genotyping are in Supplementary Table 1.

*Housing*. Animals were group-housed with free access to food and water in controlled temperature conditions (room temperature controlled at 21–22 °C, humidity between 40 and 50%), and exposed to a conventional 12-h light/dark cycle. Experiments were performed on embryos at embryonic (E) days E11.5–17.5, neonate pups at postnatal day 2–8 (P2–8), and adult (P30–56) animals of either sex. All procedures were approved by the Ethical Committee CEEA-005 Charles Darwin (authorization 26763-2020022718161012) and conducted in accordance with EU Directive 2010/63/EU. All efforts were made to reduce animal suffering and minimize the number of animals, in compliance with all relevant ethical regulations for animal testing and research.

**Viral vectors for tracing, optogenetic, and photometry experiments**. For anterograde tracing from Peri5$^{Atoh1}$ and IRT$^{Phox2b}$, we injected unilaterally 250 nl of a Cre-dependent AAV2/8-hSyn-FLEX-mGFP-2A-Synaptophysin-mRuby (Titer: $1.3 \times 10^{12}$ viral genomes (vg)/ml, Viral Core Facility Charité).

For retrograde transsynaptic tracing from muscles, we injected unilaterally 50 to 100 nl of a 1:1 viral cocktail comprised of RV-B19-ΔG-mCherry or RV-B19-ΔG-GFP (titer: $1.3 \times 10^9$ and $5.8 \times 10^8$ TU/ml respectively, Viral Vector Core—Salk Institute for Biological Studies) and an HSV-hCMV-YFP-TVA-B19G (titer: $3 \times 10^8$ TU/ml, Viral Core MIT McGovern Institute).

For retrograde tracing from IRT$^{Phox2b}$ we injected unilaterally 250 nl of a Cre-dependent AAV1/2-Syn-flex-nGToG-WPRE3 (titer: $8.1 \times 10^{11}$ vg/ml, Viral Core Facility Charité). Two weeks later we injected EnvA-RV-B19-ΔG-mCherry (titer: $3.1 \times 10^8$ vg/ml, Viral Vector Core, Salk Institute for Biological Studies).

For optogenetic and photometry experiments we respectively injected 250 nl of AAV1/2-Ef1a-DIO-stCoChR-P2A-mScarlet (titer: $3 \times 10^{13}$ vg/ml, kind gift from O. Yzhar) or 250 nl of AAV1-syn-FLEX-jGCaMP7s-WPRE (titer: $1 \times 10^{12}$ vg/ml Addgene #104487-AAV1).

## Surgical procedures

*Stereotaxic injections and implants*. All surgeries were conducted under aseptic conditions using a small animal digital stereotaxic instrument (David Kopf Instruments). Mice were anesthetized with isoflurane (3.5% at 1 l/min for induction and 2–3% at 0.3 l/min for maintenance). Buprenorphine (0.025 mg/kg) was administered subcutaneously for analgesia before surgery. A feed-back-controlled heating pad was used to maintain the animal temperature at 36 °C. Anesthetized animals were placed in a stereotaxic frame (Kopf), a 100 μl injection of lidocaine (2%) was made under the skin covering the skull, after which a small incision was made in the scalp and burr-free holes were drilled in the skull to expose the brain surface at the appropriate stereotaxic coordinates [anterior-posterior (AP) and med

ial-lateral (ML) relative to bregma; dorsal-ventral (DV) relative to brain surface at coordinate (in mm)]: −4.9 AP, 1.2 ML, 4.0 DV to target the Peri5$^{Atoh1}$ neurons; −6.7 AP, 0.5 ML, 4.2 DV to target the IRT$^{Phox2b}$ neurons. A 0.5 ML coordinate was selected for virus deliveries to the IRT$^{Phox2b}$ to circumvent the potential infection of nTS neurons along the injecting pipette track, a 4.0 DV coordinate was selected for virus deliveries to the Peri5$^{Atoh1}$ to target the center of Mo5. Viral vectors were delivered using glass micropipettes (tip diameter ca. 100 μm) backfilled with mineral oil connected to a pump (Legato 130, KD Scientific, Phymep, France) via a custom-made plunger (Phymep, France). The injector tip was lowered an additional 0.1 mm below the target site and then raised back to the target coordinate before infusion started (flow of 25 nl/min) to restrict virus diffusion to the site of

injection and prevent leakages along the needle track. After infusion, the injection pipette was maintained in position for 10 min, then raised by 100 µm increments to retract the pipet from the brain. For optogenetic and photometry experiments, 200 µm core optic fibers (0.39 NA and 0.57 NA, respectively) (Smart Laser Co., Ltd) were implanted following vector injections, ~500 µm above the sites of interest ($-4.9$ AP, 1.2 ML, 3.0 DV for $Peri5^{Atoh1}$; $-6.7$ AP, 0.9 ML, 3.6 DV for $IRT^{Phox2b}$). The optic fibers were secured via a ceramic ferrule to the skull by light-cured dental adhesive cement (Tetric Evoflow, Ivoclar Vivadent). Mice recovered from anesthesia on a heating pad before being placed, and monitored daily, in individual cages.

*Intramuscular injections.* All surgeries were conducted under aseptic conditions on P2 neonates anesthetized by deep hypothermia. For induction, pups were placed in latex sleeves gently buried in crushed ice for 3–5 min and maintenance (up to 15 min) was achieved by placing anesthetized pups on a cold pack (3–4 °C). Following small incisions of the skin to expose the targeted muscles, the viral cocktail (or 0.5% Cholera toxin subunit B (CTB) (List Labs) for labeling of the infrahyoids) was injected via a pneumatic dispense system (Picospritzer) connected to a glass pipette (tip diameter ca. 0.1 mm) mounted on a 3D micromanipulator to guide insertion in the desired muscle. Typically, 5–10 pressure pulses (100 ms, 3–5 bars) were delivered while the muscular filling was checked visually by the spreading of Fast-Green (0.025%) added to the viral solution. The pipette was withdrawn and the incision irrigated with physiological saline and closed using a 10-0 gage suture (Ethilon). The mouse was placed on a heating pad for recovery and returned to the mother. Six days postinjection (4 days for CTB), pups were deeply anesthetized, transcardially perfused with 4% paraformaldehyde (PFA) in phosphate-buffered saline (PBS), and the brains was dissected out and postfixed overnight in 4% PFA, cryoprotected in 15% sucrose in PBS and stored at $-80$ °C.

## Histology
*Immunofluorescence.* Depending on the stage, the brain was analyzed in whole embryos dissected out of the uterine horns up to E16.5, dissected out from decapitated embryos from E17.5 to P0, or after P0, dissected in cold PBS from euthanized animals perfused with cold PBS followed by 4% paraformaldehyde. Brains or embryos were postfixed in 4% paraformaldehyde overnight at 4 °C, rinsed in PBS, and cryoprotected in 15% sucrose overnight at 4 °C. Tissues were then frozen in the Tissue-Tek® OCT compound for cryo-sectioning (14–30 µm) on a CM3050s cryostat (Leica). Sections were washed for 1 h in PBS and incubated in blocking solution (5% calf serum in 0.5% Triton-X100 PBS) containing the primary antibody, applied to the surface of each slide (300 µl per slide) placed in a humidified chamber on a rotating platform. Incubation was for 4–8 h at room temperature followed by 4 °C overnight. Sections were washed in PBS ($3 \times 10$ min), then incubated with the secondary antibody in blocking solution for 2 h at room temperature, then washed in PBS ($3 \times 10$ min), air-dried, and mounted under a coverslip with fluorescence-mounting medium (Dako). Primary antibodies used were: goat anti-*Phox2b* (RD system AF4940, diluted 1:100), rabbit anti-peripherin (Abcam ab4666, 1:1000), guinea pig anti-Lmx1b (Müller et al., 2002, 1:1000), goat anti-ChAT (Millipore AB144p), 1:100), chicken anti-βGal (Abcam, ab9361, 1:1000), chicken anti-*GFP* (Aves Labs, GFP-1020,1:1000), goat anti-ChAT (Millipore, AB144p, 1:100), rabbit anti-*GFP* (Invitrogen, A11122, 1:1000), rabbit anti-*Phox2b* (Pattyn et al., 1997,1:500), rat anti-RFP (Chromotek, 5F8, 1:1000), and goat anti-CTB (List Labs, #703, 1:500). All secondary antibodies were used at 1:500 dilution: donkey anti-chicken 488 (Jackson laboratories, 703-545-155), donkey anti-chicken Cy5 (Jackson laboratories, 703-176-155), donkey anti-goat Cy5 (Jackson laboratories, 705-606-147), donkey anti-rabbit 488 (Jackson laboratories, 711-545-152) donkey anti-rabbit Cy5 (Jackson laboratories, 712-165-153), donkey anti-rat Cy3 (Jackson laboratories, 711-495-152), and donkey anti-Guinea pig Cy3 (Jackson laboratories, 706-165-148). Epifluorescence images were acquired with a NanoZoomer S210 digital slide scanner (Hamamatsu Photonics) with NDPview2+; and confocal images with a Leica SP5 confocal microscope (Leica) with Leica Application suite X. Pseudocoloring, image brightness, and contrast were adjusted using Adobe Photoshop and FIJI.

*In situ hybridization and immunohistochemistry.* For the *Atoh1* probe, primers containing SP6 and T7 overhangs were used to amplify a 607 bp region from a plasmid containing the full-length *Atoh1* CDS. The purified amplicon was then used as the template for antisense probe synthesis with T7 RNA polymerase using the following primers: Forward Primer: 5′-CGATTTTAGGTGACACTATAGAAA TCAA-CGCTCTGTCGGAGTT-3′; Reverse Primer: 5′-CTAATACGACTCACTA TAGGGACAGAGGAAGGGGAT-TGGAAGAG-3′. To generate the *Cited1* probe, a 687 bp fragment of the murine *Cited1* gene was amplified from E13.5 mouse brain cDNA (superscript III kit, Invitrogen) and cloned into pGEM-T vector (Promega), using the following primers: Forward Primer: 5′-TGGGGGGCTTAAG AGCCCGG-3′; Reverse Primer: 5′-AGGTGAGGGGTAGGATGCAG-3′. *pGEM* clones were linearized with NotI and transcribed with SP6 or T7 RNA polymerase using the DIG RNA labeling Kit (Roche 1277073) to generate antisense or sense probes. In situ hybridization was performed on 14 µm thick cryo-sections. Sections were washed for 10 min in PBS prepared in DEPC-treated water, then washed in RIPA buffer (150 mm NaCl, 1% NP-40, 0.5% Na-deoxycholate, 0.1% SDS, 1 mM EDTA, 50 mM Tris, pH 8.0) for 20 min, postfixed in 4% paraformaldehyde for

15 min followed by rinses in PBS ($3 \times 10$ min). Whenever ISH was to be followed by an immunohistochemical reaction, slides were incubated for 30 min in a mixture of 100% ethanol and 0.5% $H_2O_2$, washed in PBS ($3 \times 10$ min), then incubated in Triethanolamine containing 0.25% acetic acid for 15 min and washed again in PBS ($3 \times 10$ min). Antisense RNA probes were diluted in 200 µl hybridization buffer (5 x SSC, 10% dextran sulfate, 500 µg/mL Herring sperm DNA, 250 µg/mL Yeast-RNA, 50% formamide) and denatured at 95 °C for 5 min, cooled briefly on ice, then diluted at 100–200 ng/ml in 17 ml hybridization buffer for incubation in slide mailers, at 70 °C overnight. The next day, slides were washed for 1 h at 70 °C in 2 X SSC, 50% formamide, and 0.1% Tween 20 and for 1 h in 0.2 X SSC at 70 °C. Slides were washed in B1 buffer (0.1 M Maleic acid; pH 7.5, 0.15 M NaCl, 0.1% Tween 20), $3 \times 10$ min. The sections were then blocked for 1 h at room temperature by incubation in blocking buffer (B1 buffer supplemented with 10% heat-inactivated fetal calf serum). The blocking solution was replaced by an alkaline phosphatase-conjugated anti-DIG antibody (Roche diagnostics, 11093274910) diluted 1:200 in the blocking buffer, and sections were incubated overnight at 4 °C under a coverslip. The following day the slides were rinsed in B1 buffer ($3 \times 10$ min), equilibrated with B3 buffer (0.1 M Tris pH 9.5, 0.1 M NaCl, 50 mM $MgCl_2$, 0.1% Tween 20) for 30 min and colorimetric detection of the digoxigenin-labeled probe was performed with NBT-BCIP substrate for alkaline phosphatase (Thermo Scientific). The reaction was stopped by washing the slides in PBS-0.1% Tween 20 ($2 \times 5$ min) and fixing in 4% paraformaldehyde for 15 min. Sections were then washed in PBS-0.1% Tween 20 for 5 min each. Sections were incubated in blocking buffer (10% fetal calf serum diluted in 0.1% Tween 20 in PBS) for 1 h at room temperature, then in blocking buffer containing the primary antibody at 4 °C overnight. The next day, slides were washed for 10 min and biotinylated secondary antibody (diluted at 1:200 in blocking buffer) was applied for 2 h at room temperature and peroxidase enzyme detection of biotinylated antibody was carried out as per manufacturer's guidelines with the Vectastain Elite ABC kits (PK-6101 and PK-4005; Vector Laboratories), followed by color development using 3, 3′-Dia-minobenzidine (SIGMA FAST D4293-50SET). The reaction was stopped by washing the slides for $2 \times 5$ min in Milli-Q water, then sections were allowed to air-dry completely before mounting with Aquatex (Sigma Aldrich) for microscopy. Hybridized sections were imaged with a Leica DFC420C camera mounted on a Leica DM5500B microscope.

## Data analysis of histology
*Counts of premotor neurons and Lmx1b neurons.* Cells expressing *mCherry* and/or *nlsLacZ* were counted in a spheroid of fixed dimension and position delimiting the ipsilateral dorsal IRt, drawn on the approximately seven alternate sections that were in register with the compact formation of MoA; $n = 4$ animals, $87 \pm 20$ SEM premotor neurons per animal.

Cells expressing *Phox2b* and/or *Lmx1b* were counted as above from one side; $n = 3$ animals, $1321 \pm 46$ SEM neurons per animal.

*Inputs to IRt$^{Phox2b}$.* Labeled neurons were manually annotated as IRt seed neurons (GFP + mCherry+) or monosynaptic input neurons (mCherry+) in ImageJ. The annotated sections were aligned to the Allen Brain Atlas using QuickNII (https://www.nitrc.org/projects/quicknii) transforming the annotations into Allen Brain Atlas coordinates and corresponding Allen Brain Atlas brain structures were identified using CellfHelp https://doi.org/10.5281/zenodo.5508650. Data from individual replicates were tabulated, normalized, and pooled to generate a list of brain regions that provide monosynaptic input to IRt$^{Phox2b}$. The bar graph excludes any input below 0.3%.

## Behavioral experiments
*Timing and training.* All behavioral experiments started 4 weeks after the viral injection. Two weeks after surgery animals were habituated to head-fixation through sessions of increasing duration (2 min) every other day, starting at 2 min on day 0 and a final duration of 10 min on day 4 which corresponded to the duration of recording sessions. Animals were given condensed milk as a reward after each session. Animals used for photometry experiments were introduced to a lick port during habituation. During acquisition or manipulation animals were head-fixed within a 5 cm tube, illuminated from below and above by an LED light. Animals were water-deprived for 12 h prior to photometry experiments.

*Optogenetics.* For optogenetic photostimulation of stCoChR expressing neurons, fiber-optic cannulae were connected to a 473-nm DPSS laser (CNI, Changchun, China) through a patch cable (200 µm, 0.37 NA) and a zirconia mating sleeve (Thorlabs). Laser output was controlled using a pulse generator (accupulser, WPI), which delivered single continuous light pulses of 50–1000 ms or trains of 100 ms pulses at 4 Hz (33% duty cycle), 6 Hz (50% duty cycle), and 7 Hz (67% duty cycle). Light output through the optical fibers was adjusted to ~5 mW at the fiber tip using a digital power meter (PM100USB, Thorlabs). All light stimuli were separated by minimal periods of 10 s. Laser output was digitized at 1 kHz by a NI USB-6008 card (National Instruments) and acquired using a custom-written software package (Elphy by G Saddoc, https://www.unic.cnrs.fr/software.html).

*Photometry.* For photometry experiments, a single site fiber photometry system (Doric Lenses Inc, Canada) was used to measure the excited isosbestic (405 nm) and calcium-dependent fluorescence of jGCaMP7s (465 nm). Doric neuroscience studio software system (Doric Lenses Inc, Canada) was used to operate the photometry hardware and acquire the photometry signal. Briefly, using the "lock in mode" function, 465 and 405 nm LEDs were sinusoidally modulated at 208.616 and 572.205 Hz, respectfully (to avoid any electrical system harmonics at 50/60, 100/120, and 200/240 Hz) at an intensity of 30 μW and coupled to a patch cable (diam. 200 μm, 0.57 nA) after passing through an optical assembly (iLFMC4, Doric Lenses Inc, Canada). The modulated excitation signal was then directed through an implanted fiber-optic cannula (diam. 200 μm, 0.57 NA) onto the IRt via the mated patch cable and the emitted signal was then returned via the same patch cable to a fluorescence detector head, mounted on the optical assembly and amplified. The raw detected signal was acquired at 12 kHz and then demodulated in real time to reconstitute the excited isosbestic (405 nm) and calcium-dependent GCaMP (465 nm) signals. Contact between the tongue and the lick port during spontaneous licking bouts were registered via an SEN-1204 capacitance sensor (Sparkfun) connected to the Arduino Uno R3 microcontroller board (Arduino) and acquired at 12 kHz via the Doric fiber photometry console.

*Automated markless pose estimation.* Spontaneous and light-evoked licking sequences were filmed at portrait (Fig. 4a) and profile angles (Fig. 4d) with a CMOS camera (Jai GO-2400-C-USB) synchronized by a 5 V TTL pulse. The acquired frames (800 × 800 pixels, 120 fps,) were streamed to a hard disk using 2ndlook software (IO Industries) and compressed using a MPEG-4 codec. Portrait views were used for video tracking of optogenetically-evoked oromotor movements, while profile views were preferably used for photometry experiments, to optimize detection of the tongue, which was partially obscured by the nearby lick port when filmed from the portrait angle.

Using DeepLabCut (version 2.0.7[71]), we trained 2 ResNet-50 based neural networks to identify the tip of the tongue and lower jaw from portrait and landscape views (Fig. 4a, b, d). The "portrait" network was trained on a set of 264 frames (800 × 800 pixels) derived from 11 videos of six different mice for >400,000 iterations, reporting a train error of 1.85 pixels and test error of 6.79 pixels upon evaluation. The "profile" network was trained on a set of 90 frames (800 × 800 pixels) from four videos of four different mice for >800,000 iterations reporting a train error of 1.66 pixels and a test error of 4.57 pixels upon evaluation. These networks were then used to generate Cartesian estimates for the Y-axis position of the jaw and tongue for experimental videos.

### Data analysis

*Fiber photometry.* We analyzed behavioral and fiber photometry data using custom-written Python scripts (Python version 3.7, Python Software Foundation). Fiber photometry and photostimulation data were resampled to 120 Hz to match the acquisition rate of video recordings. Fiber photometry and photostimulation data were resampled to 120 Hz to match the acquisition rate of video recordings. Photometry data were first processed by applying a low-pass filter (Butterworth) to the calcium-dependent 465 nm and isosbestic 405 nm signals with a 20 Hz cut-off. The 465 nm signal was then normalized using the function $\Delta F/F = (F-F0)/F0$, in which F is the 465 nm signal, and F0 is the least-squared mean fit of the 405 nm signal. For each recording session in one animal, correlations between lick port contact and calcium signals were computed for all possible shifts at 120 Hz spanning from −10 to +10 s, producing one curve per session (Fig. 3g). A null correlation curve per recording session was constructed by performing the same computation after shuffling the lick port contact (Fig. 3g). All recording sessions and all null correlation curves were averaged for each animal, to produce a single mean shifted correlation curve and a null mean correlation curve per animal (Fig. S4). The maxima values of both shifted and null mean curves were retrieved for each animal ($n = 4$). A paired $t$-test between these values indicated a shifted correlation between both signals.

*Normalization of jaw and tongue pose estimation.* Cartesian pixel estimates of the jaw and tongue were corrected to a 5 mm scale bar within the video frame and smoothed using the Savitzky-Golay filter. For optogenetic experiments, the jaw position was normalized to its averaged location 50–100 ms prior to stimulation. For fiber photometry experiments, the jaw position was normalized to its average location during quiescent periods 1–3 s long. As the tongue was only present during stimulation of IRt^Phox2b^ or spontaneous lapping, we normalized the tongue distance empirically by observing the first detected instance of tongue protrusion that succeeded jaw-opening events. All positional estimates of the tongue that had a probability <5%[71] were then set to the empirically determined baseline to filter out aberrant estimates of the tongue position during periods of the recording where it was not visible.

*Licking frequency.* For data collected during optogenetic experiments, we first obtained the onsets of each lick during the 1000 ms stimulation window. These onsets were identified as the peaks of the first derive of each lick within a lick bout. Lick frequency was then calculated as the number of lick events divided by length of time from the last lick to the first lick within a lick bout. For data collected during fiber photometry, lick frequency was determined by the number of contact

events of the capacitance sensor divided by the length of time from the last to the first lick within a lick bout.

*Statistical analysis.* All data are reported as mean ± s.e.m (shaded area). *P* values for independent samples comparison were performed using a two-tailed Student's t-test.

*Statistics and reproducibility.* For physiological experiments, the number of experiments is indicated in the legends of the relevant figure. Tracing experiments and histological analyses were reproduced a minimum of three times.

**Reporting Summary**. Further information on research design is available in the Nature Research Reporting Summary linked to this article.

## Data availability
The data that support the findings of this study can be found in the Source Data provided with the paper. Microscopy data are available from the corresponding author upon reasonable request. Data from the Allen Brain Atlas was used in this study. Source data are provided with this paper.

## Code availability
All code for this paper can be found at the following address: https://doi.org/10.5281/zenodo.5508650

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

## Acknowledgements

We thank the animal facility of IBENS, the imaging facility of IBENS (supported by grants from Fédération pour la Recherche sur le Cerveau, Région Ile-de-France DIM NeRF (2009 and 2011), and France-BioImaging), Ofer Yitzar for the *AAV1-EF1a-DIO-stCoChR-P2A-mScarlet* vector. The mouse *RosaFTLG* mutant line was established at the Institut Clinique de la Souris (Phenomin-ICS) in the Genetic Engineering and Model Validation Department. Funding is from CNRS, École Normale Supérieure, INSERM, Association Nationale pour la Recherche ANR -15-CE16-0013 (to J.-F.B.), Association Nationale pour la Recherche ANR-17-CE16-0006 (to J.-F.B.), and ANR-19-CE16-0029 (to G.F.), Fondation pour la Recherche Médicale DEQ2000326472 (to J.-F.B.), "Investissements d'Avenir" program ANR-10-LABX-54 MEMO LIFE and ANR-11-IDEX-0001–02 PSL Research University), and Région Ile-de-France (to S.S.).

## Author contributions

Conceptualization: B.D., C.G., G.F. and J.-F.B. Investigation: B.D., S.S., P.B., E.R.H., Z.C., S.D. and S.A. Formal Analysis: S.M., H.C., P.B., B.D. and A.G. Supervision: C.G., G.F., J.-F.B. and J.F.A.P. Resources: C.B. Writing: B.D., G.F. and J.-F.B.

## Competing interests

The authors declare no competing interests.
