## [Peer Review File · Nature Communications]

Reviewers' Comments:

Reviewer #1:

Remarks to the Author:

This study uses anatomical, developmental and functional approaches to identify brainstem circuits involved in the control of rhythmic orofacial movement. With elegant genetic circuit tracing the authors identify two brainstem nuclei – in the intermediate reticular formation (IRT) and around the trigeminal nucleus - that are premotor to most of the muscle controlling jaw and tongue muscles. They hypothesize that this anatomical substrate might be a brainstem central pattern generator for feeding. When stimulated these two sites cause either jaw opening or jaw opening and lapping reproducing some of the movements needed for feeding. The neurons in IRT show correlated activity during natural lapping. The main conclusion of the study is that it has identified a licking CPG in the brainstem.

The anatomical and genetic dissection of the of orofacial premotor circuits is extremely elegant and well carried out using an impressive number of single and double conditional mice together with viral approaches. The results are convincing and the data support the claims about the details of the network structure. I am very enthusiastic about the story but have some problems with the text and the strong statements about the rhythmogenic potential of the network.

A.

1) The introduction. I find that the introduction does not properly introduce what the authors are aiming at. The general intro to the complexity of the brainstem and the need for molecular markers etc is appropriate but there is no real Intro to the problem of the orofacial CPG – what is known and expected from it and why Phoxb2 is connected to this function. I suggest to include such a description to make the study more appealing to a broader audience.

2) The text is generally compact. I am looking for more text to explain the methods used and some of the results and findings.

For example there is limited help to the reader to follow and understand the complicated crosses and the anatomy. The authors used two double conditional mice one in Fig. S1A and one in Fig. 1A but without a clear explanation of the first (and why it is needed). It is not described probably how cells become tdT, receive only GFP terminals or are just Chat positive MNs in Fig. 1B. It will be useful to have the MN pools attached to a function so the reader can understand what they are used for.

Fig. 1 c is essential and needs to be described in more detail. Fig S2 should be in the main text so we get the combined picture for the tracing and rationale connected with it (some sort of number should be included in the text).

Transcriptional signature: It is obvious why Atoh1 needs to be brought in – but not that the Atoh1/Phox2b population is representative for back labelled Peri5 neurons. I think it needs a better explanation. How do we know that it is not the Atoh1-/Phoxb2+ neurons that are the interesting ones? It is not clear to this reader why we need to know that IRTPhoxb2 and nTS are two structures related to lineage but with distinct molecular identities and why we need this information.

Reciprocal pattern: the paradigm for revealing the reciprocal connections between IRT and Peri5 could be better explained.

3) I think that it is too strong a statement that the functional experiments show that that IRTPhoxb2 must be the long-hypothesised licking rhythm generator. It is very difficult to show this since stimulation only produces parts of the licking behaviour and that in this set up it cannot be excluded that these cells act on other cells in IRT or elsewhere in the brainstem that generate the rhythm. In a strict sense this claim should have been supported by transsynaptic ChR2 labelling experiments from the muscles to show that pre-motor neurons indeed can trigger the rhythm. Such experiments are very difficult and not requested. But in the absence of such evidence the authors should use more careful wording also taking into account that it is discussed whether a rhythm-generating circuit is premotor or pre-premotor (see e.g. in the respiratory system).

(Feldman and others) and in the mammalian spinal cord (Kiehn, Gosgnach, McCrea, Dougherty etc).

4) The calcium imaging is nice but does only show correlation and does not add further evidence to the role as rhythm generation. Perhaps these data could have been analyzed more extensively?

5) Line 222: why will the interconnectivity support a rhythm generating role

6) Discussion

I suggest that the authors bring their findings into a general picture about the need for a licking CPG and what they have found before entering the discussion about pre-motor and coordination of movement and rhythm generation. The latter issue would need more of a background for the reader to understand the point and to carefully consider the options – also with reference to other systems (see above).

The last part of the discussion contains mainly speculations about the functional role of the transcriptional landscape and seems like an unnecessary appendix.

Reviewer #2:

Remarks to the Author:

This is a well-structured work that defines two populations of neurons (IRtPhox2b and Peri5Atho1) implicated in the control of bona fide consummatory oro-facial movements. Of the two the most interesting appears to be the IRtPhox2b population, hence its slightly more detailed functional characterization. The highlight of the work is without doubt the onto/genetic dissection of these neuronal populations and the anatomical characterization of their input/output with viral techniques. Their functional characterization is less thorough but appropriate (see caveats below) to support the key claims of the paper. From a conceptual ground, the work doesn't hugely alter our understanding of the function of the reticular formation in the control of orofacial movements. Also, in terms of the circuitry involved in orofacial control (both jaw and tongue), other works already pointed at the IRt, NST and supratrigeminal region (the Peri5) (perhaps in greater details) (see for example Takatoh et al., *elife* 2021). However, the genetic dissection of the responsible neuronal populations is, potentially (see caveats below), a significant step forward in our understanding of this circuit, whose relevance might become more evident as future works will begin to assess the descending volitional control on these populations, which is made possible by this study.

Major points:

- As far as I understand the authors used a single continuous light pulse of varying duration (100-1000ms). These are highly non-physiological conditions. What would happen if the authors stimulated with frequencies likely closer to the actual firing rate of these neurons (e.g. 5-10Hz for 50-200ms)?

- Concerning the relevance of having identified a specific IRt population responsible for jaw/tongue movements, this point would be made clearer by comparing the results of the optogenetic stimulation and of the fibre photometry between the Phox2b population and the general IRt Vglut2 (excitatory) population. Is there any difference between the two?

Minor points:

- Cholinergic neurons co-expressing glutamate in IRt have been previously implicated in orofacial movements (e.g. swallowing) (Summan Toor et al, *J Neurosci.* 2019), are VGlut2 positive Phox2b neurons also cholinergic?

- The introduction and discussion seem to lack of a fair description of the state of the art of the field with respect to circuitry and function of the brainstem control of orofacial movements (e.g.

what is already known and how exactly does this work move the field forward)

Reviewer #3:

Remarks to the Author:

The study is predicated on the assumption that the early developmental fate of lower brainstem neurons and the transcription factors implicated at this stage is a major determinant of their mature physiological role. This approach, usually combined with the clever exploitation of other population-defining markers (vesicular transporters, receptors etc.) has been especially successful in the hands of the present investigators in defining a group of brainstem neurons with a specialized role in central respiratory chemoreception (retrotrapezoid nucleus) and by others in defining functional subgroups of serotonergic or respiratory-rhythm neurons. Here the authors focus on two Phox2b-dependent neuronal clusters that they had identified in prior studies. The first cluster is a ring of neurons that surround the trigeminal motor nucleus, a region already suspected to harbor trigeminal premotor neurons. The peritrigeminal ring is also *atoh-1* dependent which distinguishes it from the motor nucleus itself and allowed the authors to manipulate the interneuronal ring selectively with intersectional genetic approaches. Thus authors were therefore able to determine the connectivity and physiological function of this ring of neurons. The results are extremely convincing.

The second focus of this study is a group of Phox2b-derived neurons located in the IRt (intermediate reticular formation). These Phox2b-derived neurons could be selectively accessed and transduced based on their stereotaxic location in Phox2b-Cre mice. The IRt is an extraordinarily complex portion of the medullary reticular formation formerly believed to be primarily implicated in autonomic regulations. As shown here this region also plays a key role in the control of orofacial movements and may contain rhythm generator.

This is an important and technically impressive study describing very novel findings regarding the genesis of orofacial movements implicated in drinking.

Reviewer #1 (Remarks to the Author):

This study uses anatomical, developmental and functional approaches to identify brainstem circuits involved in the control of rhythmic orofacial movement. With elegant genetic circuit tracing the authors identify two brainstem nuclei – in the intermediate reticular formation (IRt) and around the trigeminal nucleus - that are premotor to most of the muscle controlling jaw and tongue muscles. They hypothesize that this anatomical substrate might be a brainstem central pattern generator for feeding. When stimulated these two sites cause either jaw opening or jaw opening and lapping reproducing some of the movements needed for feeding. The neurons in IRt show correlated activity during natural lapping. The main conclusion of the study is that it has identified a licking CPG in the brainstem.

The anatomical and genetic dissection of the of orofacial premotor circuits is extremely elegant and well carried out using an impressive number of single and double conditional mice together with viral approaches. The results are convincing and the data support the claims about the details of the network structure. I am very enthusiastic about the story but have some problems with the text and the strong statements about the rhythmogenic potential of the network.

*1) The introduction. I find that the introduction does not properly introduce what the authors are aiming at. The general intro to the complexity of the brainstem and the need for molecular markers etc is appropriate but there is no real Intro to the problem of the orofacial CPG – what is known and expected from it and why *Phoxb2* is connected to this function. I suggest to include such a description to make the study more appealing to a broader audience.*

We did not aim at finding an orofacial CPG, we rather stumbled on a likely CPG while exploring the function of unknown, but genetically defined, interneurons in the hindbrain. The introduction currently reflects this. To ameliorate the introduction along the lines suggested by the referee, we have now added two phrases in bold below, a reference to Takatoh et al 2021 (at the suggestion of referee #2) and we end the introduction, as per a more classical format, with a few sentences which sum up our findings (main changes in bold):

Over decades, the reticular formation has slowly emerged from “localizatory nihilism”², and regions defined by stereotaxy [e.g.³], or cell groups defined by their projections [e.g.⁴] have been implicated in a variety of roles: **premotor neurons to orofacial or respiratory muscles^{5,6}, and — underpinning the sophisticated residual behaviors observed in decerebrate animals⁷ — rhythm and pattern generators for chewing, whisking, breathing and sighing^{3,8,9,10,11,5}. Licking is another rhythmic behavior for which a hindbrain rhythm generator is predicted¹² although the evidence is mostly extrapolated from chewing, the two behaviors possibly sharing some substrate⁹.**

However, the parsing of the reticular formation [...]

[...] thus “visceral” indeed. **To this broadened picture of the visceral nervous system, in charge of vital functions and maintenance of the interior milieu, we now add two groups of *Phox2b* interneurons, located in the reticular formation of the hindbrain, that are premotor to orofacial muscles and can command licking or lapping, a rhythmic feeding behavior essential for the intake of liquids in many terrestrial vertebrates.**

2) The text is generally compact. I am looking for more text to explain the methods used and some of the results and findings. For example, there is limited help to the reader to follow and understand the complicated crosses and the anatomy. The authors used two double conditional mice, one in Fig. S1A and one in Fig. 1A but without a clear explanation of the first (and why it is needed). It is not described probably how cells become tdT, receive only GFP terminals or are just Chat positive MNs in Fig. 1B

To remedy this problem, we have now extensively modified the technical explanations as follows, making things more explicit and splitting long sentences into shorter ones:

The vast majority of these neurons are glutamatergic, thus express the glutamate vesicular transporter *Vglut2* as shown by expression of the *Cre* and *Flpo*-dependent reporter *RC::Fela* in a *Pbox2b::Flpo;Vglut2::Cre* background (**Fig. S1A**). We used this neurotransmitter phenotype to implement an intersectional strategy that excludes the potentially confounding widespread projections of other *Pbox2b*⁺ neurons, in the locus coeruleus²¹, which are noradrenergic. We designed a novel intersectional allele (*Rosa*^{FRTTomato-loxSypGFP} or *Rosa*^{FTLG}) (**Fig. 1A**) which expresses one of two fluorophores, exclusively: action of flippase (*FLPo*) will trigger cytoplasmic expression of *tdTomato* (*tdT*), while additional action of *Cre* recombinase, will extinguish *tdT* in the cell soma and switch on instead a fusion of synaptophysin with *GFP* (*Syp-GFP*) transported to pre-synaptic sites²². When *FLPo* was driven by the *Pbox2b* promoter, and *Cre* by the *Vglut2* promoter, i.e. in *Pbox2b::Flpo;Vglut2::Cre;Rosa*^{FTLG} pups at P4, *tdT* was expressed, as expected, in the soma of the singly recombined motoneurons (which are *Pbox2b*⁺, but not glutamatergic), but lost from the doubly recombined interneurons (which are *Pbox2b*⁺ and glutamatergic) (**Fig. S1B**). The latter, in turn, had switched on *Syp-GFP*⁺ in their synaptic boutons, which covered remarkably discrete structures of the hindbrain (**Fig. S1B, Fig. 1B**), among which motor nuclei (whose function will be discussed later) featured prominently:

It will be useful to have the MN pools attached a function so the reader can understand what they are used for.

At this point in the text we would rather keep the attention on the simple notion that many *Phox2b* interneurons are premotor. Moreover, at this stage of the narrative, some motor nuclei are irrelevant to the rest of the study (MoA or Mo6, targeted by *Phox2b* premotor neurons, but not located in IRT or peri5). The role of the relevant nuclei is expounded later, as an introduction to the functional experiments, and repeating them would be awkward. To address the concern of the referee we have now added “(whose function will be discussed later)”, as cited above.

Fig. 1 c is essential and need to be described in more detail.

We have now reformulated the description as follows:

We injected a G-defective rabies virus variant encoding the fluorophore *mCherry*²⁴ together with a helper virus encoding G and the fluorophore *YFP* (*HSV-YFP-G*) in the posterior belly of the digastric muscle (**Fig. 1C**) (a jaw-abductor), known to be innervated by *Acc7*^{25,26}. Predictably, the only seed neurons (i.e. that co-express the rabies virus encoded *mCherry* and the helper virus encoded *YFP*) were found in *Acc7* (right panel in **Fig. 1C**). Premotor neurons, presynaptic to the seed motoneurons, (i.e. that express only the rabies virus encoded *mCherry*) and which, in addition, were *Pbox2b*⁺, were found at two sites only:

Fig S2 should be in the main text so we get the combined picture for the tracing and rationale connected with it (some sort of n number should be included in the text).

We respectfully decline to execute this change. Figure S2 is huge, cannot be combined with Figure 2, which would have to be reconfigured. Moreover, we do not provide the same detail in figure S2 than in Fig2, forcing us to an acrobatic formatting exercise, disproportionate to the benefit. Finally, Figure S2 contains parts which are merely contextual (i.e. the unrelated premotor landscape of laryngeal and masseter).

Transcriptional signature: It is obvious why Atob1 need to be brought in – but not that the Atob1/Phox2b population is representative for back labelled Peri5 neurons. I think it needs a better explanation. How do we know that it is not the Atob1-/Phox2b+ neurons that are the interesting ones?

The fact that neurons back-labeled from the posterior digastric INCLUDE *Atob1/Phox2b* neurons in the peri5 region is shown in **Fig. S2**. However, we do not claim that they are representative of anything else other than themselves. It is almost certain that other neurons in the same region, which are *Phox2b* but not *Atob1*, have different roles. In this paper, we are just limited, for technical reasons, to optogenetically manipulate the *Atob1/Phox2b* ones.

It is not clear to this reader why we need to know that IRTPhox2b and nTS are two structures related to lineage but with distinct molecular identities and why we need this information.

We agree that most physiologists will not need this information. However, it is an integral part of our genetic characterization of a new neuronal population, and this information might turn out helpful to guide further genetic dissections of these neurons (for example using single cell transcriptomics) or to target or interpret future optogenetic or chemogenetic manipulations.

Reciprocal pattern: the paradigm for revealing the reciprocal connections between IRT and Peri5 could be better explained.

We have now reformulated this point as follows:

In addition, anterograde tracing from IRT^{Phox2b} in a *Phox2b::Cre* background and from Peri5^{Atob1} in a *Phox2b::flpo;Atob1::Cre* background revealed, respectively, massive projections of IRT^{Phox2b} to the peri5 region (Fig. 3H) and of Peri5^{Atob1} to the IRT region. (Fig. S3C). We could not assess the precise cellular target of the former, but those of the latter included IRT^{Phox2b} (Fig. S3D, inset), suggesting reciprocal connections of the two nuclei.

3) I think that it is too strong a statement that the functional exps. show that that IRTPhox2b must be the long-hypothesised licking rhythm generator.

The original text says “the CPG [...] or part thereof”, but see below for further softening of our conclusions, along the lines of the referee’s request.

It is very difficult to show this since stimulation only produce parts of the licking behavior...

We respectfully do not understand this remark. We think that the entire licking behavior is triggered.

...and that in this set up it cannot be excluded that these cells act on other cells in IRT or elsewhere in the brainstem that generate the rhythm. In a strict sense this claim should have been supported by transsynaptic CbR2 labelling experiments from the muscles to show that pre-motor neurons indeed can trigger the rhythm. Such experiments are very difficult and not requested. But in the absence of such evidence the authors should use more careful wording also taking into account that it is discussed whether a rhythm-generating circuits are premotor or pre-premotor (see e.g in the respiratory system (Feldman and others) and in the mammalian spinal cord (Kiehn, Gosgnach, McCreia, Dougherty etc).

Indeed, the experiment mentioned by the referee is exceedingly difficult, if not impossible, because the rabies virus would kill cells before they express enough *CbR2*. Along the line suggested by the referee, we have now added the following caveats on the diagnosis of a

CPG, together with four additional references to the authors mentioned by the referee. (Note that a discussion of the optogenetic experiments has been transported from the results to the discussion section, making the latter more complete and coherent).

In addition, one of them, $\text{IRt}^{\text{Phox2b}}$, translates a tonic stimulation into a rhythmic behavior. The most parsimonious interpretation of $\text{IRt}^{\text{Phox2b}}$ is that its neurons are bifunctional: premotor through their collateralized inputs on motor nuclei, and rhythm generators, corresponding to the hypothetical licking CPG¹¹ or at least an element thereof, in the precise region where many lick-rhythmic neurons were previously recorded^(87 for reviews). It is of note that another nearby Phox2b^+ nucleus, the RTN, has intrinsic rhythmic properties, in that case related to breathing, in the neonate^{55,14}. At this stage, though, we cannot exclude that $\text{IRt}^{\text{Phox2b}}$ contains two subtypes of neurons, one premotor and the other pre-premotor, and that it is the latter which, upon photostimulation, triggers rhythmic repetition; in other words, that $\text{IRt}^{\text{Phox2b}}$ encompasses a two (or more)-level architecture, akin to models proposed for other motor behaviors^{56,57,58,59}. This possibility is made less likely by the apparent genetic homogeneity of $\text{IRt}^{\text{Phox2b}}$, whose neurons all co-express the transcriptional signature *Phox2b/Cited1*. Finally, the possibility that the rhythm would be generated by neurons elsewhere in the brainstem (recruited by $\text{IRt}^{\text{Phox2b}}$ and feeding back on it) is constrained by the limited output of $\text{IRt}^{\text{Phox2b}}$: to motor nuclei and the peri5 region.

In addition to rhythmic tongue protrusion and jaw opening, the entrainment of a full licking cycle requires the delayed activation of antagonistic muscles, as in several “burst generator” models of the locomotor CPG (e.g.⁵⁸). One substrate for such rhythmic alternation might comprise the reciprocal projections of $\text{IRt}^{\text{Phox2b}}$ and $\text{Peri5}^{\text{Atoh1}}$ (Fig. 3H, Fig. S3C,D), the former targeting tongue protractors and the latter tongue retractors.

4) *The calcium imaging is nice but does only show correlation and does no added further evidence to the role as rhythm generation. Perhaps these data could have been analyzed more extensively?*

Fiber photometry (the current gold standard for recording calcium activity in deep brain regions) is used here to show correlation between activity of $\text{IRt}^{\text{Phox2b}}$ and licking *bouts*. The limited temporal resolution of calcium indicators make this approach inherently inadequate to capture a ~7Hz rhythm, and no amount of additional analysis will resolve this.

5) *Line 222: why will the interconnectivity support a rhythm generating role*

This sentence was an allusion to the recurrent synaptic interconnections providing positive feedback that are proposed to play a role, for example in the preBötC. This is now made more explicit as follows:

“suggesting local interconnectivity of IRt neurons, possibly related to rhythmogenesis, through recurrent synaptic connections, as hypothesized for other rhythm generating structures (Del Negro et al 2006)”

6) *Discussion*

I suggest that the authors bring their findings into a general picture about the need for a licking CPG and what they have found before entering the discussion about pre-motor and coordination of movement and rhythm generation. The latter issue would need more of a background for the reader to understand the point and to carefully consider the options – also with reference to other systems (see above).

We hope that the above-mentioned modifications to the introduction, restructuring and lengthening of the discussion on the likelihood that $\text{IRt}^{\text{Phox2b}}$ is a CPG, and how it would

mechanistically compare to other ones, plus added bibliographic references to other CPGs take care of this point.

The last part of the discussion contains mainly speculations about the functional role of the transcriptional landscape and seems like an unnecessary appendix.

The last part treats the subject of evolution of physiological functions and of neuron types, and is admittedly of no import to the physiologist. However, we believe that part of the originality of our paper is to sit at the border of several disciplines (physiology, development and evolution) and we would like to leave it that way.

Reviewer #2 (Remarks to the Author):

This is a well-structured work that defines two populations of neurons (IRt^{Phox2b} and Peri5^{Atoh1}) implicated in the control of bona fide consummatory oro-facial movements. Of the two the most interesting appears to be the IRt^{Phox2b} population, hence its slightly more detailed functional characterization. The highlight of the work is without doubt the onto/genetic dissection of these neuronal populations and the anatomical characterization of their input output with viral techniques. Their functional characterization is less thorough but appropriate (see caveats below) to support the key claims of the paper. From a conceptual ground, the work doesn't hugely alter our understanding of the function of the reticular formation in the control of orofacial movements. Also, in terms of the circuitry involved in orofacial control (both jaw and tongue), other works already pointed at the IRt, NST and supratrigeminal region (the Peri5) (perhaps in greater details) (see for example Takatoh et al., elife 2021). However, the genetic dissection of the responsible neuronal populations is, potentially (see caveats below), a significant step forward in our understanding of this circuit, whose relevance might become more evident as future works will begin to assess the descending volitional control on these populations, which is made possible by this study.

Major points:

- As far as I understand the authors used a single continuous light pulse of varying duration (100-1000ms). These are highly non-physiological conditions. What would happen if the authors stimulated with frequencies likely closer the actual firing rate of these neurons (e.g. 5-10Hz for 50-200ms)?

In our view, what the light pulse should ideally emulate is the firing pattern of input structures to the IRt. This pattern is unknown and, for all we know, could be a tonic drive (for example from the cortex, as already mentioned in the text), in which case constant illumination might be a decent approximation, after all.

This, said, as requested by the referee, we have now added 100ms stimulations delivered at 4, 6 and 7Hz and we show that we analogically entrain the lapping movements. This is commented in the main text as:

[...] while IRt^{Phox2b} but not Peri5^{Atoh1} can protract the tongue, in line with the targeting of hypoglossal motoneurons for tongue protractors by the former and tongue retractors by the latter (**Fig. 3D,E,J,K**). Delivering the stimulus at 4, 5 or 7Hz led to an analogical repetition of the movement (**Fig. S4A**) showing a lack of refractory period in that frequency range. Lengthening the light pulse [...]

And illustrated as a new panel A in **Fig. S4**:

- Concerning the relevance of having identified a specific IRt population responsible for jaw/tongue movements, this point would be made clearer by comparing the results of the optogenetic stimulation and of the fibre photometry between the *Phox2b* population and the general IRt *Vglut2* (excitatory) population. Is there any difference between the two?

This experiment, which would require redoing all the physiological experiments in a different genetic background, is an enormous task, and in all likelihood, would only confirm (by triggering more complex or incoherent movements) that *Phox2b* is a more specific marker than *Vglut2*, i.e. that *Phox2b::Cre* background selects a subset of neurons. For one thing, it is difficult to restrict tracing to just neurons in the IRt because *vglut2* expression is seamless between the IRt and surrounding regions.

Moreover (and the referee's remark highlights a potentially misleading aspect of our terminology) although the name IRt^{*Phox2b*} could suggest that the IRt (or even part of it), is a spatially defined region homogeneously made of *Phox2b* neurons, this is not the case. To make this point clearer we have added evidence that IRt^{*Phox2b*} neurons are intermingled with *Phox2b*-negative cells, which are also glutamatergic. This now appears in the text as follows:

Unlike the nTS, IRt^{*Phox2b*} neurons are intermingled with glutamatergic neurons of other types (*Phox2b*-negative) (Fig. S3).

And in a new Fig. S3

Not surprisingly, tracing from IRt in a *Vglut2::Cre* background shows much wider projections than in a *Phox2b::Cre* background, as can be seen on the Allen Brain Atlas at: https://connectivity.brain-map.org/projection/experiment/302016107?imageId=302016579&initImage=TWO_PHOTON&x=17863&y=15990&z=3

Along the same lines, and although not requested by any referee, we have slightly altered the abstract to make the point of marker expression less abstract. The sentence:

“These neuronal groups, defined by unique transcriptional codes and developmental origins, IRt^{*Phox2b*} and Peri5^{*Atob1*}, are located, [...]”

Is now replaced by:

“These neuronal groups, IRt^{Phox2b} and $Peri5^{Atab1}$, are marked by expression of the pan-autonomic homeobox gene *Phox2b* and are located, [...]”

Minor points:

- *Cholinergic neurons co-expressing glutamate in IRt have been previously implicated in orofacial movements (e.g. swallowing) (Summan Toor et al, J Neurosci. 2019), are VGlut2 positive Phox2b neurons also cholinergic?*

The *ChAT+ / Vglut2+* neurons alluded to by the referee correspond to the PiCo (Anderson et al, 2016), whose region is inhibited en masse by Toor et al (2019) by the GABA-A receptor agonist Isoguvacine, leading to deficits in swallowing. The neurons we study (IRt^{Phox2b}) are dorsal to the PiCo (which occupies the ventral IRt), but most importantly are *ChAT* negative, and the PiCo is *Phox2b*-negative, as shown on the immunofluorescence below, at low (left) and high (right) magnification.

- *The introduction and discussion seem to lack of a fair description of the state of the art of the field with respect to circuitry and function of the brainstem control of orofacial movements (e.g. what is already known and how exactly does this work move the field forward).*

In the absence of a more precise request from the referee, we respectfully doubt the appropriateness of including in this paper on a likely CPG for licking, a full review on the state of the art for “*circuitry and function of the brainstem control of orofacial movements*” (including whisking, sniffing, swallowing, chewing, gaping etc...and their disparate states of advancement). Concerning licking, the main data prior to our paper was the electrophysiological recording of lick-rhythmic neurons in IRt and PCrt, described in the review by Travers et al (1997), that we reference several times in the text. The advance (i.e. that we find cells in the same region that can command licking, and a genetic signature for them) seems clear from the current state of the text, and we find it difficult to be more assertive, especially since referee #2 asks us to be less so (see above).

In response to referee#1 (see above), we have added sentences in the introduction and discussion as well as several references, which we hope expands the context of our work to the satisfaction of referee#2.

Reviewer #3 (Remarks to the Author):

The study is predicated on the assumption that the early developmental fate of lower brainstem neurons and the

transcription factors implicated at this stage is a major determinant of their mature physiological role. This approach, usually combined with the clever exploitation of other population-defining markers (vesicular transporters, receptors etc.) has been especially successful in the hands of the present investigators in defining a group of brainstem neurons with a specialized role in central respiratory chemoreception (retrotrapezoid nucleus) and by others in defining functional subgroups of serotonergic or respiratory-rhythm neurons. Here the authors focus on two Pbox2b-dependent neuronal clusters that they had identified in prior studies. The first cluster is a ring of neurons that surround the trigeminal motor nucleus, a region already suspected to harbor trigeminal premotor neurons. The peritrigeminal ring is also atoh-1 dependent which distinguishes it from the motor nucleus itself and allowed the authors to manipulate the interneuronal ring selectively with intersectional genetic approaches. Thus authors were therefore able to determine the connectivity and physiological function of this ring of neurons. The results are extremely convincing.

The second focus of this study is a group of Pbox2b-derived neurons located in the IRT (intermediate reticular formation). These Pbox2b-derived neurons could be selectively accessed and transduced based on their stereotaxic location in Pbox2b-Cre mice. The IRT is an extraordinarily complex portion of the medullary reticular formation formerly believed to be primarily implicated in autonomic regulations. As shown here this region also plays a key role in the control of orofacial movements and may contain rhythm generator.

This is an important and technically impressive study describing very novel findings regarding the genesis of orofacial movements implicated in drinking.

We thank the referee for the positive comments.

Reviewers' Comments:

Reviewer #1:

Remarks to the Author:

I have no further comments to this manuscript.

Reviewer #2:

Remarks to the Author:

The authors have addressed all my comments/questions and I am happy to support publication, I congratulate the authors on an excellent work.

Reviewer #3:

Remarks to the Author:

Having requested no change to the first version my only comment is that the revised one seems to have addressed very aptly the points raised by the other reviewers.

*Reviewer #1 (Remarks to the Author):
I have no further comments to this manuscript.*

Thank you

*Reviewer #2 (Remarks to the Author):
The authors have addressed all my comments/ questions and I am happy to support publication, I congratulate the authors on an excellent work.*

Thank you

*Reviewer #3 (Remarks to the Author):
Having requested no change to the first version my only comment is that the revised one seems to have addressed very aptly the points raised by the other reviewers.*

Thank you very much.